# Revisiting Graph Contrastive Learning from the Perspective of Graph Spectrum

**Nian Liu**[1]**, Xiao Wang**[1,2*]**, Deyu Bo**[1]**, Chuan Shi**[1,2*]**, Jian Pei**[3]

[1]Beijing University of Posts and Telecommunications
[2]Peng Cheng Laboratory
[3]Simon Fraser University
{nianliu, xiaowang, bodeyu, shichuan}@bupt.edu.cn, jpei@cs.sfu.ca

## Abstract

Graph Contrastive Learning (GCL), learning the node representations by augmenting graphs, has attracted considerable attentions. Despite the proliferation of various graph augmentation strategies, some fundamental questions still remain unclear: what information is essentially encoded into the learned representations by GCL? Are there some general graph augmentation rules behind different augmentations? If so, what are they and what insights can they bring? In this paper, we answer these questions by establishing the connection between GCL and graph spectrum. By an experimental investigation in spectral domain, we firstly find the General grAph augMEntation (GAME) rule for GCL, i.e., the difference of the high-frequency parts between two augmented graphs should be larger than that of low-frequency parts. This rule reveals the fundamental principle to revisit the current graph augmentations and design new effective graph augmentations. Then we theoretically prove that GCL is able to learn the invariance information by contrastive invariance theorem, together with our GAME rule, for the first time, we uncover that the learned representations by GCL essentially encode the low-frequency information, which explains why GCL works. Guided by this rule, we propose a spectral graph contrastive learning module (SpCo[1]), which is a general and GCL-friendly plug-in. We combine it with different existing GCL models, and extensive experiments well demonstrate that it can further improve the performances of a wide variety of different GCL methods.

## 1 Introduction

Graph Neural Networks (GNNs) learn the node representations in a graph mainly by message passing. GNNs have attracted significant interest and found many applications [11, 23, 14]. Training the high quality GNNs heavily relies on task-specific labels, while it is well known that manually annotating nodes in graphs is costly and time-consuming [10]. Therefore, Graph Contrastive Learning (GCL) is developed as a typical technique for self-supervised learning without the explicit usage of labels [28, 9, 36].

The traditional GCL framework (Fig. 1 (a)) mainly includes three components: graph augmentation, graph representation learning by an encoder, and contrastive loss. In essence, GCL aims to maximize agreement between augmentations to learn invariant representations [36]. Typical GCL methods have

---

[*]Corresponding authors.
[1]Code available at https://github.com/liun-online/SpCo

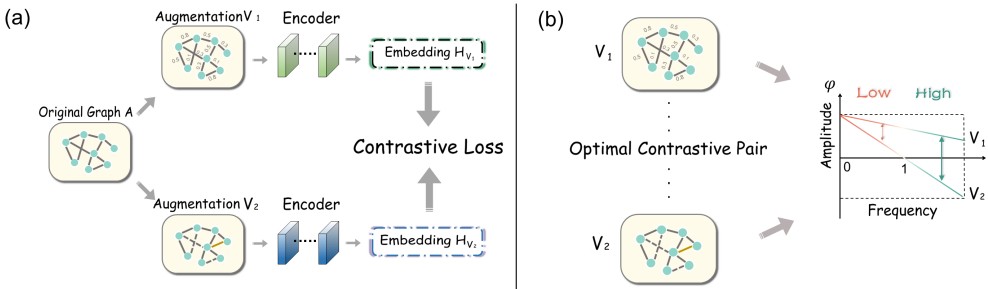

Figure 1: (a) The general framework of GCL. (b) The illustration of our findings in the empirical study (Section 3). If two contrasted graphs have a larger margin between high-frequency part than low-frequency part, they will boost the GCL. We call such two graphs as optimal contrastive pair.

sought to elaborately design different graph augmentation strategies. For example, the heuristic based methods including node or edge dropping [32], feature masking [33], and diffusion [9]; and the learning based methods including InfoMin [26, 30], disentanglement [13], and adversarial training [31]. Although various graph augmentation strategies are proposed, the fundamental augmentation mechanism is not well understood. *What information should we preserve or discard in an augmented graph? Are there some general rules across different graph augmentation strategies? How to use those general rules to validate and improve the current GCL methods?* This paper explores those questions.

Essentially, an augmented graph is obtained by changing some components in the original graph and thus strength of frequencies [20] in graph spectrum. This natural and intuitive connection between graph augmentation and graph spectrum inspires us to explore the effectiveness of augmentations from the spectral domain. We start with an empirical study (Section 3) to understand the importance of low-frequency and high-frequency information in GCL. Our findings indicate that both the lowest-frequency information and the high-frequency information are important in GCL. Retaining more high-frequency information is particularly helpful to improve the performance of GCL. However, as shown in Fig. 1 (b), the way of handling high-frequency information in two contrasted graphs $V_1$ and $V_2$ should be different, which can be finally summarized as a general graph augmentation (GAME) rule: the difference of amplitudes of high frequencies in two contrasted graphs should be larger than that of low frequencies.

To explain the GAME rule, we need to understand what information is encoded into the learned representations by GCL. We propose the contrastive invariance (Theorem 1), which, for the first time, theoretically proves that GCL can learn the invariance information from two contrasted graphs. Meanwhile, as can be seen in Fig. 1 (b), because the difference of amplitudes of lowest-frequency information is much smaller than that of high-frequency information, the lowest-frequency information will be the approximately invariant pattern between the two graphs $V_1$ and $V_2$. Therefore, with such two augmentations $V_1$ and $V_2$, we can conclude that the information learned by GCL is mainly the low-frequency information, whose usefulness has been well demonstrated [6]. This not only explains why GCL works, but also provides a clear and concise demonstration of which augmentation strategy is better, as verified by the experiments in Section 4.

Based on our findings and theoretical analysis, we define two augmentations satisfying the GAME rule are called an optimal contrastive pair. Then, we propose a novel **sp**ectral graph **co**ntrastive learning (SpCo), a general GCL framework, which can boost existing GCL methods with optimal contrastive pairs. Specifically, to ensure that the learned augmented graph is an optimal contrastive pair with the original adjacency matrix, we need to make the amplitude of its high frequency ascend while keeping the low frequency the same as the original structure. We model this process as an optimization objective based on matrix perturbation theory, which can be solved by Sinkhorn's Iteration [24] and finally obtain the augmented structure used for the following target GCL model.

Our contributions are summarized as follows. Firstly, we answer the question "what information is learned by GCL and whether there exists a general augmentation rule". To the best of our knowledge, this is the first attempt to fundamentally explore the augmentation strategies for GCL from spectral domain. We not only reveal the general graph augmentation rule behind different augmentation

strategies, but also explain why GCL works by proposing the contrastive invariance theorem. Our work provides deeper understanding on the nature of GCL. Secondly, we answer the question "how to utilize the augmentation rule for GCL". We show that the augmentation rule provides a novel insight to estimate the current augmentation strategies. We propose a novel concept optimal contrastive pair and theoretically derive a general framework SpCo, which is able to improve the performance of existing GCL methods. Last, we choose three typical GCL methods as target methods, and plug SpCo into them. We validate the effectiveness of SpCo on five datasets. We consistently gain improvements compared with those target methods.

## 2 Preliminaries

Let $\mathcal{G} = (\mathcal{V}, \xi)$ represent an undirected attributed graph, where $\mathcal{V}$ is the set of $N$ nodes and $\xi \subseteq \mathcal{V} \times \mathcal{V}$ is the set of edges. All edges formulate an adjacency matrix $\boldsymbol{A} \in \{0,1\}^{N \times N}$, where $\boldsymbol{A}_{ij} \in \{0,1\}$ denotes the relation between nodes $i$ and $j$ in $\mathcal{V}$. The node degree matrix $\boldsymbol{D} = diag(d_1, \ldots . d_n)$, where $d_i = \sum_{j \in \mathcal{V}} \boldsymbol{A}_{ij}$ is the degree of node $i \in \mathcal{V}$. Graph $\mathcal{G}$ is often associated with a node feature matrix $\boldsymbol{X} = [x_1, x_2, \ldots, x_N] \in \mathbb{R}^{N \times d}$, where $x_i$ is a $d$ dimensional feature vector of node $i \in \mathcal{V}$. Let $\mathcal{L} = \boldsymbol{D} - \boldsymbol{A}$ be the unnormalized graph Laplacian of $\mathcal{G}$. If we set symmetric normalized adjacency matrix as $\hat{\boldsymbol{A}} = \boldsymbol{D}^{-\frac{1}{2}} \boldsymbol{A} \boldsymbol{D}^{-\frac{1}{2}}$, then $\hat{\mathcal{L}} = \boldsymbol{I_n} - \hat{\boldsymbol{A}} = \boldsymbol{D}^{-\frac{1}{2}}(\boldsymbol{D} - \boldsymbol{A})\boldsymbol{D}^{-\frac{1}{2}}$ is the symmetric normalized graph Laplacian.

Since $\hat{\mathcal{L}}$ is symmetric normalized, its eigen-decomposition is $\boldsymbol{U}\boldsymbol{\Lambda}\boldsymbol{U}^{\top}$, where $\boldsymbol{\Lambda} = diag(\lambda_1, \ldots, \lambda_N)$ and $\boldsymbol{U} = [\boldsymbol{u_1}^{\top}, \ldots, \boldsymbol{u_N}^{\top}] \in \mathbb{R}^{N \times N}$ are the eigenvalues and eigenvectors of $\hat{\mathcal{L}}$, respectively. Without loss of generality, assume $0 \leq \lambda_1 \leq \cdots \leq \lambda_N < 2$ (where we approximate $\lambda_N \approx 2$ [11]). Denote by $\mathcal{F}_{\mathcal{L}} = \{\lambda_1, \ldots, \lambda_{\lfloor N/2 \rfloor}\}$ the amplitudes of *low-frequency components* and by $\mathcal{F}_{\mathcal{H}} = \{\lambda_{\lfloor N/2 \rfloor + 1}, \ldots, \lambda_N\}$ the amplitudes of *high-frequency components*. The **graph spectrum** is defined as these amplitudes of different frequency components, denoted as $\phi(\lambda)$, indicating which parts of frequency are enhanced or weakened [20]. Additionally, we rewrite $\hat{\mathcal{L}} = \lambda_1 \cdot \boldsymbol{u_1}\boldsymbol{u_1}^{\top} + \cdots + \lambda_N \cdot \boldsymbol{u_N}\boldsymbol{u_N}^{\top}$, where we define term $\boldsymbol{u_i}\boldsymbol{u_i}^{\top} \in \mathbb{R}^{N \times N}$ as the eigenspace related to $\lambda_i$, denoted as $\boldsymbol{S_i}$.

**Graph Contrastive Learning** (GCL) [28, 9, 33] aims to learn discriminative embeddings without supervision, whose pipeline is shown in Fig. 1(a). We summarize the representative GCL in Appendix E. Specifically, two augmentations are randomly extracted from $\boldsymbol{A}$ in a predefined way and are encoded by GCN [11] to obtain the node embeddings under these two augmentations. Then, for one target node, its embedding in one augmentation is learned to be close to the embeddings of its positive samples in the other augmentation and be far away from those of its negative samples. Models built in this way are capable of discriminating similar nodes from dissimilar ones. For example, some graph contrastive methods [36, 32, 35] use classical InfoNCE loss [19] as the optimization objective:

$$\mathcal{L}(\boldsymbol{h}_i^{\boldsymbol{V_1}}, \boldsymbol{h}_i^{\boldsymbol{V_2}}) = \log \frac{\exp(\theta(\boldsymbol{h}_i^{\boldsymbol{V_1}}, \boldsymbol{h}_i^{\boldsymbol{V_2}})/\tau)}{\exp(\theta(\boldsymbol{h}_i^{\boldsymbol{V_1}}, \boldsymbol{h}_i^{\boldsymbol{V_2}})/\tau) + \sum_{k \neq i} \exp(\theta(\boldsymbol{h}_i^{\boldsymbol{V_1}}, \boldsymbol{h}_k^{\boldsymbol{V_2}})/\tau)}, \quad (1)$$

where $\boldsymbol{h}_i^{\boldsymbol{V_1}}$ and $\boldsymbol{h}_i^{\boldsymbol{V_2}}$ are the embeddings of node $i$ under augmentations $V_1$ and $V_2$, respectively, $\theta$ is the similarity metric, such as cosine similarity, and $\tau$ is a temperature parameter. The total loss is $\mathcal{L}_{InfoNCE} = \sum_i \frac{1}{2} \left( \mathcal{L}(\boldsymbol{h}_i^{\boldsymbol{V_1}}, \boldsymbol{h}_i^{\boldsymbol{V_2}}) + \mathcal{L}(\boldsymbol{h}_i^{\boldsymbol{V_2}}, \boldsymbol{h}_i^{\boldsymbol{V_1}}) \right)$.

## 3 Impact of Graph Augmentation: An Experimental Investigation

In this section, we aim to explore what information should be considered in two contrasted augmentations from the perspective of graph spectrum. Specifically, we design a simple GCL framework shown in Fig. 2. Two input augmentations are adjacency matrix $\boldsymbol{A}$ and generated $\boldsymbol{V}$. Then, we utilize a shared GCN with one layer as the encoder to encode $\boldsymbol{A}$ and $\boldsymbol{V}$ and get their nodes embeddings as $\boldsymbol{H_A}$ and $\boldsymbol{H_V}$. We train the GCN by utilizing InfoNCE loss as in Eq. (1). More experimental settings can be found in appendix B.1.

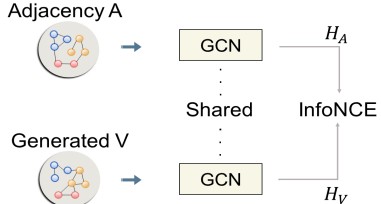

Figure 2: The case study model.

**Generating augmentation** $V$ We construct the augmented graph by extracting information with different frequencies from the original graph, so that we can analyze the effect of different information. This process is shown in Fig. 3. Specifically, we divide the eigenvalues of $\mathcal{L}$ into $\mathcal{F_L}$ and $\mathcal{F_H}$ parts, and conduct augmentations in these two parts, respectively. Taking the augmentation in $\mathcal{F_L}$ for example, we keep the high-frequency part as $\boldsymbol{u_{N/2}u_{N/2}^\top} + \cdots + \boldsymbol{u_N u_N^\top}$. Then, we gradually add the

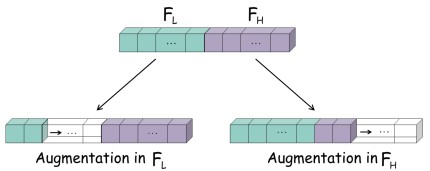

Figure 3: The generation of $\boldsymbol{V}$.

eigenspaces in $\mathcal{F_L}$ back with rates $[20\%, 40\%, 60\%, 80\%]$, starting from the lowest frequency. Therefore, $\boldsymbol{V}$ augmenting 20% in $\mathcal{F_L}$ is $\boldsymbol{u_1 u_1^\top} + \cdots + \boldsymbol{u_{0.2*N/2}u_{0.2*N/2}^\top} + \boldsymbol{u_{N/2}u_{N/2}^\top} + \cdots + \boldsymbol{u_N u_N^\top}$. Similarly, $\boldsymbol{V}$ augmenting 20% in $\mathcal{F_H}$ is $\boldsymbol{u_1 u_1^\top} + \cdots + \boldsymbol{u_{N/2}u_{N/2}^\top} + \boldsymbol{u_{(N+1)/2}u_{(N+1)/2}^\top} + \cdots + \boldsymbol{u_{0.7N}u_{0.7N}^\top}$. Please note that we set graph spectrum of $\boldsymbol{V}$, $\phi_{\boldsymbol{V}}(\lambda) = 1, \forall \lambda \in [0,2]$ above, in that we just want to test the effect of different $\boldsymbol{u_i u_i^\top}$ and avoid the influence from eigenvalues $\lambda$ [18].

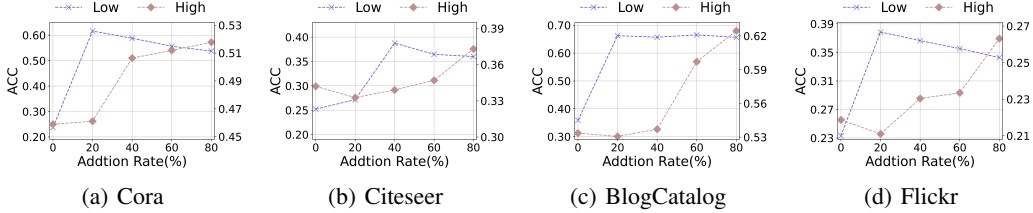

| (a) Cora | (b) Citeseer | (c) BlogCatalog | (d) Flickr |

Figure 4: The results of case study on four datasets. The x-axis means different addition rate of different frequency interval, and y-axis means the performance on ACC. The performance of augmentations in $\mathcal{F_L}$ are plotted on the left y-axis, and in $\mathcal{F_H}$ are plotted on the right y-axis.

**Results and analyses** We conduct the node classification on four datasets: Cora, Citeseer [11], BlogCatalog, and Flickr [16]. The accuracy (ACC) is shown in Fig. 4. In appendix B.2, we also report the results when both high and low frequency components are added back in the high-to-low frequency order. **Results.** For each dataset, in generated $\boldsymbol{V}$, (1) when the lowest part of frequencies are kept, the best performance is achieved; (2) when more frequencies in $\mathcal{F_H}$ are involved, the performance generally rises. **Analyses.** From the graph spectra of $\boldsymbol{A}$ and $\boldsymbol{V}$ shown in Fig. 5, we can see that in generated $\boldsymbol{V}$, (1) when the lowest part of frequencies are kept, the difference of amplitude, i.e., the graph spectrum, in $\mathcal{F_L}$ between $\boldsymbol{A}$ and $\boldsymbol{V}$ becomes smaller; (2) when more frequencies in $\mathcal{F_H}$ are involved, the margin of graph spectrum in $\mathcal{F_H}$ between $\boldsymbol{A}$ and $\boldsymbol{V}$

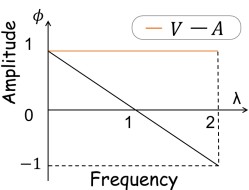

Figure 5: The spectrum of $\boldsymbol{A}$ and $\boldsymbol{V}$.

becomes larger. Combining results and observations, we propose the following general **G**raph **AugME**ntation rule, called GAME rule[1]:

---

**The General Graph Augmentation Rule**

Given two random augmentations $\boldsymbol{V_1}$ and $\boldsymbol{V_2}$, their graph spectra are $\phi_{\boldsymbol{V_1}}(\lambda)$ and $\phi_{\boldsymbol{V_2}}(\lambda)$. Then, $\forall \lambda_m \in [1,2]$ and $\lambda_n \in [0,1]$, $\boldsymbol{V_1}$ and $\boldsymbol{V_2}$ are an effective pair of graph augmentations if the following condition is satisfied:
$$|\phi_{\boldsymbol{V_1}}(\lambda_m) - \phi_{\boldsymbol{V_2}}(\lambda_m)| > |\phi_{\boldsymbol{V_1}}(\lambda_n) - \phi_{\boldsymbol{V_2}}(\lambda_n)|.$$
We define such pair of augmentations as optimal contrastive pair.

---

[1]Although this rule is derived from contrasting $\boldsymbol{A}$ and $\boldsymbol{V}$, the selection of certain views does not curb the generality of GAME rule. Considering that most of augmentations are obtained from the raw adjacency matrix $\boldsymbol{A}$, it is a natural setting that one view is fixed as $\boldsymbol{A}$ and the other is an augmented one.

# 4 Analysis of The General Graph Augmentation Rule

In this section, we aim to verify the correctness of GAME rule that whether two contrasted augmentations satisfying GAME rule can perform better in downstream tasks from experimental and theoretical analysis.

**Experimental analysis** We substitute existing augmentations proposed by MVGRL [9], GCA [36] and GraphCL [32] for augmentation $V$ in the case. Specifically, MVGRL proposes PPR matrix, heat diffusion matrix and pair-wise distance matrix. GCA mainly randomly drops edges based on Degree, Eigenvector and PageRank. GraphCL adopts random node dropping, edge perturbation and subgraph sampling. The nine augmentations almost cover the mainstream augmentations in GCL. To accurately depict the change of the amplitude after these augmentations for some $\lambda_i$, we turn to matrix perturbation theory [1] [25]:

$$\Delta\lambda_i = \lambda_i' - \lambda_i = \boldsymbol{u}_i^\top \Delta \boldsymbol{A} \boldsymbol{u}_i - \lambda_i \boldsymbol{u}_i^\top \Delta \boldsymbol{D} \boldsymbol{u}_i + \mathcal{O}(||\Delta\boldsymbol{A}||), \tag{2}$$

where $\lambda_i'$ is the eigenvalue after change, $\Delta\boldsymbol{A} = \boldsymbol{A}' - \boldsymbol{A}$ represent the modification of edges after augmentation, and $\Delta\boldsymbol{D}$ is the respective change in degree matrix. With Eq. (2), we calculate the eigenvalues on Cora after each augmentation, and plot their graph spectra in Fig. 6. Simultaneously, we use the GCL framework in Section 3 to separately contrast adjacency matrix $\boldsymbol{A}$ and these augmentations, and results are shown in Table 1. As shown in Fig. 6, PPR matrix, Heat diffusion matrix and Distance matrix better accord with GAME rule, where they have small difference with $\boldsymbol{A}$ in $\mathcal{F}_\mathcal{L}$, and have a large difference in $\mathcal{F}_\mathcal{H}$. Therefore, they outperform other augmentations in Table 1.

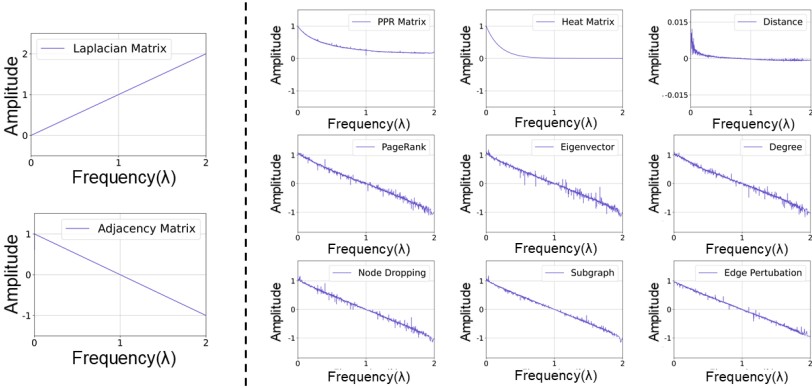

Figure 6: The graph spectra of laplacian, adjacency matrix and nine existing augmentations.

Table 1: Performance of different existing augmentations to verify the GAME rule.

| Methods | GraphCL | | | GCA | | | MVGRL | | |
|---|---|---|---|---|---|---|---|---|---|
| Type | Subgraph | Node dropping | Edge perturbation | Degree | PageRank | Eigenvector | PPR | Heat | Distance |
| Results | 34.9±3.5 | 29.8±2.3 | 37.7±4.4 | 40.2±4.1 | 38.5±5.0 | 42.1±4.9 | **58.0**±1.6 | **49.9**±4.2 | **46.1**±7.5 |

We also test the GAME rule in another circumstance, where we contrast among three cases: $\boldsymbol{A}$ and $\boldsymbol{A}^2$ (two-hop of $\boldsymbol{A}$), $\boldsymbol{A}$ and $\boldsymbol{A}$, and $\boldsymbol{A}^2$ and $\boldsymbol{A}^2$. The results are given in Appendix C.

**Theoretical analysis** We have the following theorem to depict the learning process of the GCL.

**Theorem 1.** *(Contrastive Invariance) Given adjacency matrix $\boldsymbol{A}$ and the generated augmentation $\boldsymbol{V}$, the amplitudes of $i$-th frequency of $\boldsymbol{A}$ and $\boldsymbol{V}$ are $\lambda_i$ and $\gamma_i$, respectively. With the optimization of InfoNCE loss $\mathcal{L}_{InfoNCE}$, the following upper bound is established:*

$$\mathcal{L}_{InfoNCE} \leq \frac{1+N}{2} \sum_i \theta_i \left[ 2 - (\lambda_i - \gamma_i)^2 \right],$$

*where $\theta_i$ is an adaptive weight of the $i$th term.*

---

[1] Here, we do not use eigenvalue decomposition to obtain $\lambda'$ of $\boldsymbol{A}'$, because the obtained $\lambda'$ are unordered compared with previous $\lambda$ of $\boldsymbol{A}$. That is to say, for certain $\lambda_i$ of $\boldsymbol{A}$, we cannot figure out which eigenvalue of $\boldsymbol{A}'$ matches to it after decomposition, so we cannot calculate the change $\Delta\lambda_i$ for $\lambda_i$ in this case.

The proof is given in the Appendix A.1, where we simplify GCN without the activation function. Theorem 1 indicates an upper bound of GCL loss, implying that maximizing the contrastive loss equals to maximize the upper bound. So, larger $\theta_i$ will be assigned to the smaller $(\lambda_i - \gamma_i)^2$, or $\lambda_i \approx \gamma_i$. Meanwhile, if $\lambda_i \approx \gamma_i$, these two contrasted augmentations are regarded to share the invariance at $i$th frequency. Therefore, with contrastive learning, the encoder will emphasize the invariance between two contrasted augmentations from spectrum domain. To our best knowledge, theorem 1, for the first time, theoretically proves that GCL can capture the invariance between two augmentations. Please recall that GAME rule suggests that the difference between two augmentations in $\mathcal{F}_{\mathcal{L}}$ is smaller. Thus, under the guidance of GAME rule, GCL attempts to capture the common low-frequency information of two augmentations. Thus, GAME rule points out a general augmentation strategy to manipulate encoder to capture low-frequency information, which achieves a better performance.

# 5 Spectral Graph Contrastive Learning

Based on the GAME Rule, we mainly aim to learn a general and GCL-friendly transformation $\Delta_{\boldsymbol{A}}$ from adjacency matrix $\boldsymbol{A}$ to a new augmentation $\boldsymbol{A}_{\_}$ (or $\Delta_{\boldsymbol{A}} = \boldsymbol{A}_{\_} - \boldsymbol{A}$), where $\boldsymbol{A}$ and $\boldsymbol{A}_{\_}$ are required to be an optimal contrastive pair. Then, they are fed into existing GCL method $\Phi$, i.e. augmenting with the same strategies of $\Phi$ to generate $\boldsymbol{V_1}$ and $\boldsymbol{V_2}$ and training with the corresponding contrastive loss, shown in Fig. 7. The whole pipeline is our proposed spectral graph contrastive learning (SpCo), which can boost existing GCL methods.

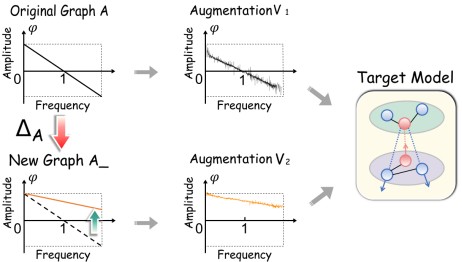

Figure 7: Combine SpCo with existing GCL.

Firstly, we separate $\Delta \boldsymbol{A} = \Delta_{\boldsymbol{A}+} - \Delta_{\boldsymbol{A}-}$, where $\Delta_{\boldsymbol{A}+}$ and $\Delta_{\boldsymbol{A}-}$ indicate which edge is added and deleted, respectively. Next, we indicate how to learn $\Delta_{\boldsymbol{A}+}$, while the calculation of $\Delta_{\boldsymbol{A}-}$ is similar. Based on our theoretical derivation in Appendix. A.2, the following optimization objective of $\Delta_{\boldsymbol{A}+}$ should be maximized:

$$\mathcal{J} = \underbrace{< \mathcal{C}, \ \Delta_{\boldsymbol{A}+} >^2}_{\text{Matching Term}} + \underbrace{\epsilon H(\Delta_{\boldsymbol{A}+})}_{\text{Entropy Reg.}} + \underbrace{< \boldsymbol{f}, \Delta_{\boldsymbol{A}+} \mathbb{1}_n - \boldsymbol{a} > + < \boldsymbol{g}, \Delta_{\boldsymbol{A}+}^{\top} \mathbb{1}_n - \boldsymbol{b} >}_{\text{Lagrange Constraint Conditions}}, \quad (3)$$

This objective consists of three components: (1) **Matching Term**. $\forall \boldsymbol{P}, \boldsymbol{Q} \in \mathbb{R}^{N \times N}, < \boldsymbol{P}, \boldsymbol{Q} >= \sum_{ij} \boldsymbol{P}_{ij} \boldsymbol{Q}_{ij}$. To maximize $< \mathcal{C}, \ \Delta_{\boldsymbol{A}+} >^2$, $\Delta_{\boldsymbol{A}+}$ should learn to "match" or be similar to $\mathcal{C}$. In A.2, we define $\mathcal{C} = \boldsymbol{U}g(\lambda)\boldsymbol{U}^{\top}$, where $\boldsymbol{U}$ and $g(\lambda)$ are eigenvector matrix and some function about eigenvalues of $\boldsymbol{A}$. According to GAME rule, we set $\phi_{\Delta}(\lambda) = |\phi_{\boldsymbol{A}}(\lambda) - \phi_{\boldsymbol{A}\_}(\lambda)|$, and we need $\phi_{\Delta}(\lambda_m) > \phi_{\Delta}(\lambda_n), \forall \lambda_m \in [1,2]$ and $\lambda_n \in [0,1]$. Therefore, we stipulate that $\phi_{\Delta}(\lambda)$ should be a monotone increasing function. Since $\mathcal{C}$ will guide $\Delta_{\boldsymbol{A}+}$ to capture the change of difference between graph spectra ($\phi_{\Delta}(\lambda)$), we naturally set $g(\lambda)$ of $\mathcal{C}$ also a monotone increasing function. Furthermore, we notice that the graph spectrum of Laplacian $\mathcal{L}$ does meet our need about $g(\lambda)$ (shown in Fig. 6), so we simply set $\mathcal{C} = \Theta \mathcal{L}$, where $\Theta$ is a parameter updating in training. (2) **Entropy Regularization**. Here, $H(\boldsymbol{P}) = -\sum_{i,j} \boldsymbol{P}_{i,j}(\log(\boldsymbol{P}_{i,j}) - 1)$ [21], and $\epsilon$ is the weight of this term. This term aims to increase the uncertainty of the learnt $\Delta_{\boldsymbol{A}+}$, which encourages more edges (entries in $\Delta_{\boldsymbol{A}+}$) to join in optimization. (3) **Lagrange Constraint Conditions**. $\boldsymbol{f} \in \mathbb{R}^{N \times 1}$ and $\boldsymbol{g} \in \mathbb{R}^{N \times 1}$ are Lagrange multipliers, and $\boldsymbol{a} \in \mathbb{R}^{N \times 1}$ and $\boldsymbol{b} \in \mathbb{R}^{N \times 1}$ are distributions[1]. This term restrains the row and column sums of $\Delta_{\boldsymbol{A}+}$ within some limitation.

Next, we expound how to solve eq. (3). The partial of $\mathcal{J}$ with respect to $\Delta_{\boldsymbol{A}+}$ is as following:

$$\partial \mathcal{J} / \partial (\Delta_{\boldsymbol{A}+})_{ij} = 2 < \mathcal{C}, \ \Delta_{\boldsymbol{A}+} > \mathcal{C}_{ij} - \epsilon \log(\Delta_{\boldsymbol{A}+})_{ij} + \boldsymbol{f}_i + \boldsymbol{g}_j \quad (4a)$$

$$= m_{ij} + 2\mathcal{C}_{ij}^2(\Delta_{\boldsymbol{A}+})_{ij} - \epsilon \log(\Delta_{\boldsymbol{A}+})_{ij} + \boldsymbol{f}_i + \boldsymbol{g}_j, \quad (4b)$$

where we separate $2\mathcal{C}_{ij}^2(\Delta_{\boldsymbol{A}+})_{ij}$ from $2 < \mathcal{C}, \ \Delta_{\boldsymbol{A}+} > \mathcal{C}_{ij}$, and set the rest part as $m_{ij}$. The next theorem points out when $\mathcal{J}$ can get the maximal value in the domain of definition $(\Delta_{\boldsymbol{A}+})_{ij} \in (0,1)$:

---

[1] We define $\boldsymbol{a}$ and $\boldsymbol{b}$ are both node degree distribution in this paper.

**Theorem 2.** *Given $(\Delta_{A+})_{ij} \in (0,1)$, $\mathcal{J}$ exists the maximal value, iff*
*(1) $\mathcal{C}_{ij}^2 < -\frac{f_i + g_j + m_{ij}}{2}$, and $f_i + g_j + m_{ij} < 0$, or*
*(2) $\frac{\epsilon}{2} < \mathcal{C}_{ij}^2 < \frac{\epsilon}{2} \exp(-\frac{f_i + g_j + m_{ij} + \epsilon}{2})$, and $f_i + g_j + m_{ij} + \epsilon < 0$.*

We provide the proof in the Appendix A.3. Normally, we should let eq. (4b) equal to zero and get the analytical solution of $(\Delta_{A+})_{ij}$. However, eq. (4b) is a transcendental equation because of the coexistence of linear term and logarithm. Thus, we require eq. (4a) to equal to 0. As the training goes on, $\Delta_{A+}$ does not change sharply. So, we firstly rewrite eq. (4a) as follows:

$$\partial \mathcal{J} / \partial(\Delta_{A+})_{ij} \approx 2 < \mathcal{C}, \Delta'_{A+} > \mathcal{C}_{ij} - \epsilon \log(\Delta_{A+})_{ij} + f_i + g_j. \tag{5}$$

Compared with eq. (4a), eq. (5) only replaces $\Delta_{A+}$ with $\Delta'_{A+}$ in the first term, where $\Delta'_{A+}$ is obtained from the last training epoch, and frozen at the current epoch. In this case, the matrix form of solution of current epoch is:

$$\Delta_{A+} = diag(\boldsymbol{u}) \exp\left(2 < \mathcal{C}, \Delta'_{A+} > \mathcal{C} / \epsilon\right) diag(\boldsymbol{v}) = \boldsymbol{U}_+ \boldsymbol{K}_+ \boldsymbol{V}_+, \tag{6}$$

where $\boldsymbol{U}_+ = diag(\boldsymbol{u}_i) = diag\left(\exp\left(\frac{f_i}{\epsilon}\right)\right)$ and $\boldsymbol{V}_+ = diag(\boldsymbol{v}_j) = diag\left(\exp\left(\frac{g_j}{\epsilon}\right)\right)$. To further calculate $\boldsymbol{U}_+$ and $\boldsymbol{V}_+$, we restrain the row and column sums of $\Delta_{A+}$ according to Lagrange Constraint Conditions: $\boldsymbol{u} * (\boldsymbol{K}_+ \boldsymbol{v}) = \boldsymbol{a}$ and $\boldsymbol{v} * (\boldsymbol{K}_+^\top \boldsymbol{u}) = \boldsymbol{b}$. We solve this matrix scaling problem [17] by Sinkhorn's Iteration [24], which is shown in Algorithm 1 [4]. There exists a upper bound of the difference between $\Delta_{A+}$ and $\Delta'_{A+}$:

**Theorem 3.** *After Sinkhorn's Iteration, the bound between $\Delta_{A+}$ and $\Delta'_{A+}$ is:*
$$\left|\alpha(\Delta_{A+})_{ij} - (\Delta_{A+})'_{ij}\right| \le \frac{\alpha}{\epsilon^2(1-\gamma)}\{d(r^{(0)}, \boldsymbol{a}) + d(c^{(0)}, \boldsymbol{b})\} + \alpha(1 + \frac{|m_{ij}|}{\epsilon}),$$
*where $\alpha = \frac{\epsilon}{2\mathcal{C}_{ij}^2}$. $\forall(x, x') \in (\mathbb{R}_+^n)^2$, $d(x, x')$ is the Hilbert's projective metric [3] on $\mathbb{R}_+^n$. $\gamma$ is $\kappa(\boldsymbol{K}_+)$, and $\kappa$ is contraction ratio [7]. $r^{(0)}$ and $c^{(0)}$ are the row and column sum vectors of $\boldsymbol{K}_+$.*

The proof is given in Appendix A.4. The calculation of $\Delta_{A-}$ is similar as $\Delta_{A+}$ shown as follows:

$$\Delta_{A-} = diag(\boldsymbol{u}') \exp\left(-2 < \mathcal{C}, \Delta'_{A-} > \mathcal{C} / \epsilon\right) diag(\boldsymbol{v}') = \boldsymbol{U}_- \boldsymbol{K}_- \boldsymbol{V}_-, \tag{7}$$

where $diag(\boldsymbol{u}')$, $diag(\boldsymbol{v}')$ and $\Delta'_{A-}$ have the similar meanings as in eq. (6).

Finally, we get the solution $\Delta_A = \Delta_{A+} - \Delta_{A-}$, utilizing eq. (6) and eq. (7). With learnt transformation $\Delta_A$, we can obtain the new augmentation $A_-$ as:

$$A_- = A + \eta \cdot \mathbb{S} * \Delta_A, \tag{8}$$

where '*' means element-wise product, and $\eta$ is the combination coefficient. To make $\Delta_A$ sparse, we use scope matrix $\mathbb{S}$ to limit our focus, e.g. one-hop neighbors for each node. The whole algorithm is given in Algorithm 2.

---

**Algorithm 1:** Sinkhorn's Iteration

**Input** : Matrix $\boldsymbol{K}$, distribution $\boldsymbol{a} \in \mathbb{R}^{N \times 1}$ and $\boldsymbol{b} \in \mathbb{R}^{N \times 1}$
**Params** : Iteration number $Iter$
**Output** : $\Delta_{A+}$ (or $\Delta_{A-}$)

1 Initialize $\boldsymbol{u} = [1/N, 1/N, \ldots, 1/N]_{1 \times N}$;
2 $\overline{\boldsymbol{K}} = diags(1./\boldsymbol{a})\boldsymbol{K}$;
3 **for** $i = 1$ *to* $Iter$ **do**
4     $\boldsymbol{u} = 1./\overline{\boldsymbol{K}}\left(\boldsymbol{b}/\boldsymbol{K}^\top \boldsymbol{u}\right)$;
5 **end**
6 $\boldsymbol{v} = \boldsymbol{b}/\boldsymbol{K}^\top \boldsymbol{u}$;
7 $\Delta_{A+}/\Delta_{A-} = diag(\boldsymbol{u})\boldsymbol{K} diag(\boldsymbol{v})$;
8 **return** $\Delta_{A+}/\Delta_{A-}$;

---

**Algorithm 2:** The proposed SpCo

**Input** : $\Phi$, augmentation $Aug_\Phi$, $A$, $\mathcal{L}$, $X$
**Params** : Total epochs $T$, update epochs $\Omega$, $\mathbb{S}$, $\eta$, $\Theta$, $\epsilon$, $\boldsymbol{a}$ and $\boldsymbol{b}$

1 **for** $i = 1$ *to* $T$ **do**
2     $\mathcal{C} = \Theta\mathcal{L}$;
3     Calculate $\boldsymbol{K}_+ / \boldsymbol{K}_-$ in eq. (6), (7);
4     Get $\Delta_{A+} / \Delta_{A-}$ through Algorithm 1;
5     $A_- = A + \eta(\Delta_{A+} - \Delta_{A-})\mathbb{S}$ with eq. (8);
6     Update $\Theta$;
7     **for** $j = 1$ *to* $\Omega$ **do**
8        $V_1, V_2 = Aug_\Phi(A), Aug_\Phi(A_-)$;
9        Train $\Phi(V_1, V_2, X)$;
10     **end**
11 **end**

---

# 6 Experiments

In this section, we mainly evaluate the performance of proposed SpCo on five datasets: Cora, Citeseer, Pubmed [11], BlogCatalog and Flickr [16]. Details of datasets are in Appendix D.2. We select two categories of baselines: semi-supervised GNN models {GCN [11], GAT [27]} and six representative graph contrastive learning methods {DGI [28], MVGRL [9], GRACE [35], GCA [36], GraphCL [32], CCA-SSG [33]}. These GCL methods can be divided into three categories based on their contrastive losses: BCE loss (DGI, MVGRL), InfoNCE loss (GRACE, GCA, GraphCL) and CCA loss (CCA-SSG). To verify the applicability of our SpCo, we select one baseline from each category (DGI, GRACE and CCA-SSG) to integrate with SpCo. The detailed descriptions of DGI, GRACE and CCA-SSG are given in Appendix D.6. Experimental implementation details are given in Appendix D.1.

Table 2: Quantitative results ($\%\pm\sigma$) on node classification.

| Datasets | Metrics | GCN | GAT | DGI | **DGI+SpCo** | MVGRL | GRACE | **GRACE+SpCo** | GCA | GraphCL | CCA-SSG | **CCA+SpCo** |
|---|---|---|---|---|---|---|---|---|---|---|---|---|
| Cora | Ma-F1 | 79.6±0.7 | 81.3±0.3 | 80.4±0.7 | **81.1±0.5** | 81.5±0.5 | 79.2±1.0 | **80.3±0.8** | 79.9±1.1 | 80.7±0.9 | 82.9±0.8 | **83.6±0.4** |
| | Mi-F1 | 80.7±0.6 | 82.3±0.2 | 82.0±0.5 | **82.8±0.7** | 82.8±0.4 | 80.0±1.0 | **81.2±0.9** | 81.1±1.0 | 82.3±0.9 | 83.6±0.9 | **84.3±0.4** |
| Citeseer | Ma-F1 | 68.1±0.5 | 67.5±0.2 | 67.7±0.9 | **68.3±0.5** | 66.8±0.7 | 65.1±1.2 | **65.1±0.8** | 62.8±1.3 | 67.8±1.0 | 67.9±1.0 | **68.5±1.0** |
| | Mi-F1 | 70.9±0.5 | 72.0±0.9 | 71.7±0.8 | **72.4±0.5** | 72.5±0.5 | 68.7±1.1 | **69.4±1.0** | 65.9±1.0 | 71.9±0.9 | 73.1±0.7 | **73.6±1.1** |
| BlogCatalog | Ma-F1 | 71.2±1.2 | 67.6±2.2 | 68.2±1.3 | **71.5±0.8** | 80.3±3.6 | 67.7±1.2 | **68.2±0.4** | 71.7±0.4 | 63.9±2.1 | 72.0±0.5 | **72.8±0.3** |
| | Mi-F1 | 72.1±1.3 | 68.3±2.2 | 68.8±1.4 | **72.3±0.9** | 80.9±3.6 | 68.5±1.3 | **69.4±1.3** | 72.7±0.5 | 64.6±2.1 | 73.0±0.5 | **73.7±0.3** |
| Flickr | Ma-F1 | 48.9±1.6 | 35.0±0.8 | 31.2±1.6 | **33.7±0.7** | 31.2±2.9 | 35.7±1.3 | **36.3±1.4** | 41.2±0.5 | 32.1±1.1 | 37.0±1.1 | **38.7±0.6** |
| | Mi-F1 | 50.2±1.2 | 37.1±0.3 | 33.0±1.6 | **35.2±0.7** | 33.4±3.0 | 37.3±1.0 | **38.1±1.3** | 42.2±0.6 | 34.5±0.9 | 39.3±0.9 | **40.4±0.4** |
| PubMed | Ma-F1 | 78.5±0.3 | 77.4±0.2 | 76.8±0.9 | **77.6±0.6** | 79.8±0.4 | 80.0±0.7 | **80.3±0.3** | 80.8±0.6 | 77.0±0.4 | 80.7±0.6 | **81.3±0.3** |
| | Mi-F1 | 78.9±0.3 | 77.8±0.2 | 76.7±0.9 | **77.4±0.5** | 79.7±0.3 | 79.9±0.7 | **80.7±0.2** | 81.4±0.6 | 76.8±0.5 | 81.0±0.6 | **81.5±0.4** |

## 6.1 Node classification

To more comprehensively evaluate our model, we use two common evaluation metrics, including Macro-F1 and Micro-F1. The results are reported in Table 2, where the training set contains 20 nodes per class. As can be seen, the proposed SpCo can generally improve the performances of the corresponding original models on all datasets, which verifies that our SpCo is widely applicable and effective. We also choose 5 and 10 labeled nodes per class as training set respectively, which are reported in Appendix D.4.1.

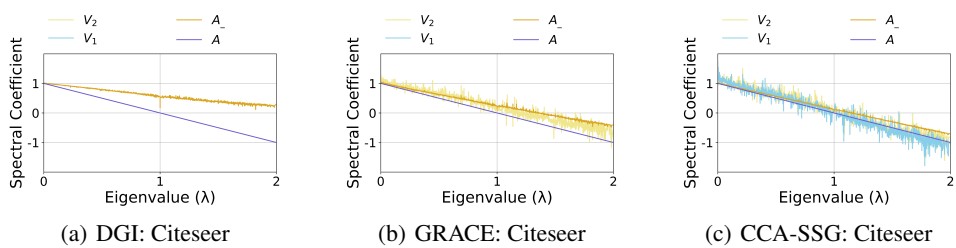

|  (a) DGI: Citeseer | (b) GRACE: Citeseer | (c) CCA-SSG: Citeseer |
|---|---|---|

Figure 8: The visualisation of graph spectrum on Citeseer.

## 6.2 Visualisation of graph spectrum

In this section, we test if the learnt view $A\_$ and $A$ meet the GAME rule. We plot the graph spectrum of $A\_$, $A$, $V_1$ and $V_2$ in one figure for each method on Citeseer, which are shown in Fig. 8. Here, we discard the impact of self-loop operation. For DGI, it does not use topological augmentation. [2] Therefore, we only plot $A\_$ and $A$ for it. For GRACE, the augmentation strength of $V_1$ is set to 0. Thus, the plot of $V_1$ is same with $A$. From the figures, we can see that the difference between $A\_$

---

[2]Although in DGI, the authors summary a vector to depict the global view, this summary vector does not reflect any graph structure, thus we think DGI does not have special augmentation strategies on topology.

and $A$ is smaller in $\mathcal{F}_{\mathcal{L}}$ than in $\mathcal{F}_{\mathcal{H}}$, which proves that they are optimal contrastive pair. Meanwhile, they can drive $V_1$ and $V_2$ also to obey the GAME rule, and thus boost the final results. More results on Cora are given in Appendix D.4.2.

## 6.3 Hyper-parameter sensitivity

In this subsection, we systematically investigate the sensitivity of two parameters: matrix $\mathcal{C}$ and $\epsilon$. We conduct node classification on Cora and BlogCatalog datasets and report the Micro-F1 values. More experiments of hyper-parameters are given in Appendix D.4.3.

**Analysis of $\mathcal{C}$.** The matrix $\mathcal{C}$ directly affects the final structures of the $\Delta_{A+}$ and $\Delta_{A-}$. Therefore, we give three kinds of $\mathcal{C}$: $\mathcal{I}+\mathcal{L}$, $\mathcal{L}$ and $\mathcal{I}+\mathcal{L}+\mathcal{L}^2$, and corresponding results are shown in Fig. 9. From the figures, we can see that $\mathcal{L}$ is the best choice compared with two candidates. So, we use $\theta\mathcal{L}$ as $\mathcal{C}$. Other well-designed $\mathcal{C}$ can also replace $\mathcal{L}$ here.

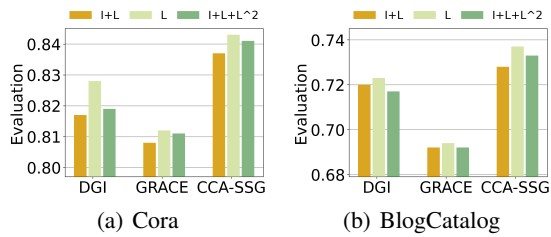

(a) Cora  (b) BlogCatalog

Figure 9: The comparison between three candidates for $\mathcal{C}$.

**Analysis of $\epsilon$.** The $\epsilon$ in eq. (3) controls the strength of entropy regularization, and in eq. (6) also controls the smoothness of exponential. We vary the value of it and plot the results on BlogCatalog in Fig. 10. From the results, we know that $\epsilon$ is a sensitive parameter for SpCo. If $\epsilon$ is too small, the effect of entropy term will diminish. And if $\epsilon$ is too large, the entropy term will interfere the molding of new structure. More results on Cora are given in Appendix D.4.3.

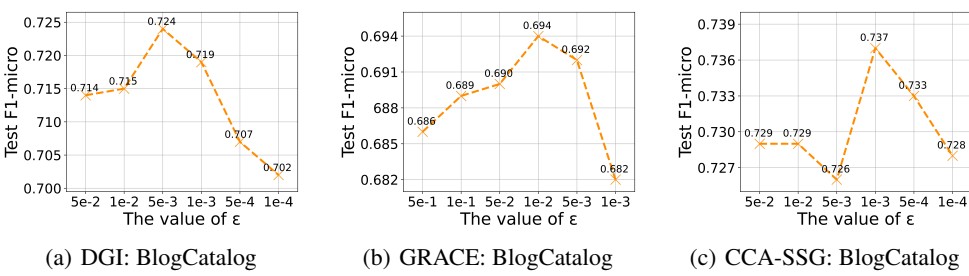

(a) DGI: BlogCatalog  (b) GRACE: BlogCatalog  (c) CCA-SSG: BlogCatalog

Figure 10: Analysis of the hyper-parameter $\epsilon$ on BlogCatalog.

## 7 Conclusion

In this paper, we fundamentally explore the topological augmentation of GCL from spectral domain. We propose contrastive invariance theorem, and discover a general augmentation (GAME) rule, which deepen our understanding of the essence of GCL. Then, we propose a general augmentation plug-in based on GAME rule, SpCo, to boost existing GCL methods. Extensive experiments verify the effectiveness of SpCo.

**Limitations and broader impact.** On potential limitation is that this work mainly focuses on the homophily graph, rather than the heterophily graphs [1], where high-frequency information is more useful. Despite the great development of GCL, some theoretical foundations are still lacking. Our work points out the great potential of graph spectrum in GCL, and may open a new path to understand and design GCL. Other than that, we do not foresee any direct negative impacts on the society.

## Acknowledgments and Disclosure of Funding

This work is supported in part by the National Natural Science Foundation of China (No. U20B2045, 62192784, 62172052, 62002029, U1936014).

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
