# A  Proof and Derivation

## A.1  Proof of Contrastive Invariance

*Proof.* We start our proof from the contrastive loss in eq. (1), rewritten as follows:

$$\mathcal{L}(\boldsymbol{h}_i^{V_1}, \boldsymbol{h}_i^{V_2}) = \log \frac{\exp(\theta(\boldsymbol{h}_i^{V_1}, \boldsymbol{h}_i^{V_2})/\tau)}{\exp(\theta(\boldsymbol{h}_i^{V_1}, \boldsymbol{h}_i^{V_2})/\tau) + \sum_{k \neq i} \exp(\theta(\boldsymbol{h}_i^{V_1}, \boldsymbol{h}_k^{V_2})/\tau)}, \tag{9}$$

Here, for simplification, we set $\tau = 1$, $\theta$ is dot product similarity, the number of GCN layer is 1 and without non-linear activation. In the case study, we utilize augmentations $\boldsymbol{A}$ and $\boldsymbol{V}$ as two inputs. And then, for the optimization of node $i$, we have:

$$\begin{aligned}
\mathcal{L}(\boldsymbol{h}_i^A, \boldsymbol{h}_i^V) &= \log \frac{\exp(\boldsymbol{h}_i^A \boldsymbol{h}_i^{V\top})}{\sum_k \exp(\boldsymbol{h}_i^A \boldsymbol{h}_k^{V\top})} \\
&= \boldsymbol{h}_i^A \boldsymbol{h}_i^{V\top} - \log \sum_k \exp(\boldsymbol{h}_i^A \boldsymbol{h}_k^{V\top}) \\
&\leq \boldsymbol{h}_i^A \boldsymbol{h}_i^{V\top} - \log N \cdot \exp\left(\frac{\sum_k \boldsymbol{h}_i^A \boldsymbol{h}_k^{V\top}}{N}\right) \tag{10a} \\
&= \boldsymbol{h}_i^A \boldsymbol{h}_i^{V\top} - \log N - \frac{1}{N}\sum_k \boldsymbol{h}_i^A \boldsymbol{h}_k^{V\top} \\
&\Rightarrow \boldsymbol{h}_i^A \boldsymbol{h}_i^{V\top} - \frac{1}{N}\sum_k \boldsymbol{h}_i^A \boldsymbol{h}_k^{V\top}. \tag{10b}
\end{aligned}$$

In formula (10a) we utilize inequality of arithmetic and geometric means, where given any N positive numbers, they meet $\frac{x_1 + \cdots + x_N}{N} \geq \sqrt[N]{x_1 \ldots x_N}$. And in formula (10b), we neglect the constant $\log N$. Then, we gather the losses of all nodes as follows:

$$\begin{aligned}
\mathcal{L} &= \sum_i \mathcal{L}(\boldsymbol{h}_i^A, \boldsymbol{h}_i^V) \\
&\leq \sum_i \boldsymbol{h}_i^A \boldsymbol{h}_i^{V\top} - \sum_i \frac{1}{N}\sum_k \boldsymbol{h}_i^A \boldsymbol{h}_k^{V\top} \tag{11} \\
&= tr(\boldsymbol{H}^A \boldsymbol{H}^{V\top}) - \frac{1}{N} sum(\boldsymbol{H}^A \boldsymbol{H}^{V\top}),
\end{aligned}$$

where $\boldsymbol{H}^A$ and $\boldsymbol{H}^{V\top}$ denote the embeddings matrices for all nodes under $\boldsymbol{A}$ and $\boldsymbol{V}$, respectively. $tr(\boldsymbol{U})$ means the trace of matrix $\boldsymbol{U}$, and $sum(\boldsymbol{U})$ means the sum of all elements in matrix $\boldsymbol{U}$. Please recall that $\boldsymbol{V}$ is composed of different eigenspaces of $\boldsymbol{A}$, so we can have that $\boldsymbol{A} = \boldsymbol{U}\boldsymbol{\Lambda}\boldsymbol{U}^\top$ and $\boldsymbol{V} = \boldsymbol{U}\boldsymbol{\Gamma}\boldsymbol{U}^\top$, where $\boldsymbol{U}$ is the collection of eigenspaces, and $\boldsymbol{\Lambda} = diag(\lambda_1, \ldots, \lambda_N)$ and $\boldsymbol{\Gamma} = diag(\gamma_1, \ldots, \gamma_N)$ are the diagonal weight matrices of different eigenspaces. Because we simplify that only one layer GCN without activation function is used, we have the following formula:

$$\boldsymbol{H}^A \boldsymbol{H}^{V\top} = \boldsymbol{A}\boldsymbol{X}\boldsymbol{W}\boldsymbol{W}\boldsymbol{X}\boldsymbol{V}, \tag{12}$$

where $\boldsymbol{W}$ is learnable parameters of the encoder. Then, we set $\boldsymbol{M} = \boldsymbol{X}\boldsymbol{W}\boldsymbol{W}\boldsymbol{X}$. And thus, we can view $\boldsymbol{M}$ as the similarities of features between every two nodes after projection. As mentioned in previous studies, linked nodes often have similar features [15], and thus their similarity should be also higher after projection. Therefore w.l.o.g., we assume that transformed features between nodes present a kind of high-order proximity, given in [34]:

**Assumption 1.** *(**High-order Proximity**)* $\boldsymbol{M} = w_0 + w_1\boldsymbol{A} + w_2\boldsymbol{A}^2 + \cdots + w_q\boldsymbol{A}^q$, where $\boldsymbol{A}^i$ means matrix multiplications between $i$ $\boldsymbol{A}$s, and $w_i$ is the weight of that term.

In other words, this assumption aims to expand $\boldsymbol{M}$ with the weighted sum of different orders of $\boldsymbol{A}$. Furthermore, we have that:

**Theorem 4.** *When $q \geq N - 1$, $\boldsymbol{M} = \boldsymbol{U}\boldsymbol{\Theta}\boldsymbol{U}^\top$, where $\Theta = diag(\theta_1, \ldots, \theta_N)$. And $\{\theta_1, \ldots, \theta_N\}$ are N different parameters, if $\{\lambda_1, \ldots, \lambda_N\}$ are N different frequency amplitudes.*

*Proof.*

$$
\begin{aligned}
\boldsymbol{M} &= w_0 + w_1\boldsymbol{A} + w_2\boldsymbol{A}^2 + \cdots + w_q\boldsymbol{A}^q \\
&= \boldsymbol{U}w_0\boldsymbol{I}\boldsymbol{U}^\top + \boldsymbol{U}w_1\boldsymbol{\Lambda}\boldsymbol{U}^\top + \boldsymbol{U}w_2\boldsymbol{\Lambda}^2\boldsymbol{U}^\top + \cdots + \boldsymbol{U}w_q\boldsymbol{\Lambda}^q\boldsymbol{U}^\top \\
&= \boldsymbol{U}[w_0\boldsymbol{I} + w_1\boldsymbol{\Lambda} + w_2\boldsymbol{\Lambda}^2 + \cdots + w_q\boldsymbol{\Lambda}^q]\boldsymbol{U}^\top \\
&= \boldsymbol{U}
\begin{bmatrix}
\sum_{i=0}^{q} w_i\lambda_1^i & & & \\
& \sum_{i=0}^{q} w_i\lambda_2^i & & \\
& & \cdots & \\
& & & \sum_{i=0}^{q} w_i\lambda_N^i
\end{bmatrix}
\boldsymbol{U}^\top
\end{aligned}
\tag{13}
$$

We firstly give the following equation set:

$$
\begin{cases}
w_0 + w_1\lambda_1 + \cdots + w_q\lambda_1^q = \theta_1 \\
w_0 + w_1\lambda_2 + \cdots + w_q\lambda_2^q = \theta_2 \\
\qquad\qquad \cdots \\
w_0 + w_1\lambda_N + \cdots + w_q\lambda_N^q = \theta_N,
\end{cases}
\tag{14}
$$

where $\{\theta_1, \ldots, \theta_N\}$ are N randomly given parameters. And the determinant of coefficient of this equation set is $\mathbb{D} = \begin{vmatrix} 1 & \lambda_1 & \cdots & \lambda_1^q \\ 1 & \lambda_2 & \cdots & \lambda_2^q \\ & & \cdots & \\ 1 & \lambda_N & \cdots & \lambda_N^q \end{vmatrix}$. From the linear algebra, we know that if (1) $q = N - 1$, $\mathbb{D}$ is the Vandermonde determinant, and then $\mathbb{D} = \prod_{N \geq i \geq j \geq 1}(\lambda_i - \lambda_j)$. If $\forall i, j \in [1, N]$, we all have $\lambda_i \neq \lambda_j$, so $\mathbb{D} \neq 0$. In this case, the coefficient matrix is a full-rank matrix, and there will be one unique solution of the set (14). If $\exists$ one pair of $i, j \in [1, N]$ and $\lambda_i = \lambda_j$, we can force $\theta_i$ and $\theta_j$ to be equal, which means that the encoder deploys the same effect on the repeated frequencies. The number of repeated frequencies is considerably smaller that that of different frequencies, so we neglect this case; (2) $q > N - 1$, the number of unknown numbers is larger than that of equations in set (14), so there are infinite solutions. Above all, theorem 4 is proved. $\square$

Theorem 4 points that $\boldsymbol{M}$ can be easily rewritten in $U\Theta U^\top$, if we use sufficient orders of $\boldsymbol{A}$ to approach $\boldsymbol{M}$. Combined with this theorem, we resume the derivation of eq. (12) as follows:

$$
\begin{aligned}
\boldsymbol{H}^{\boldsymbol{A}}\boldsymbol{H}^{\boldsymbol{V}^\top} &= \boldsymbol{A}\boldsymbol{M}\boldsymbol{V} \\
&= \boldsymbol{U}\boldsymbol{\Lambda}\boldsymbol{U}^\top\boldsymbol{M}\boldsymbol{U}\boldsymbol{\Gamma}\boldsymbol{U}^\top = \boldsymbol{U}\boldsymbol{\Lambda}\boldsymbol{\Theta}\boldsymbol{\Gamma}\boldsymbol{U} \\
&= \boldsymbol{U}
\begin{bmatrix}
\lambda_1\theta_1\gamma_1 & & & \\
& \lambda_2\theta_2\gamma_2 & & \\
& & \cdots & \\
& & & \lambda_n\theta_n\gamma_n
\end{bmatrix}
\boldsymbol{U}^\top \\
&= \lambda_1\theta_1\gamma_1\boldsymbol{u_1}\boldsymbol{u_1}^\top + \lambda_2\theta_2\gamma_2\boldsymbol{u_2}\boldsymbol{u_2}^\top + \cdots + \lambda_N\theta_N\gamma_N\boldsymbol{u_N}\boldsymbol{u_N}^\top
\end{aligned}
\tag{15}
$$

Therefore, we have:

$$
tr(\boldsymbol{H}^{\boldsymbol{A}}\boldsymbol{H}^{\boldsymbol{V}^\top}) = \sum_i \lambda_i\theta_i\gamma_i, \text{ and } sum(\boldsymbol{H}^{\boldsymbol{A}}\boldsymbol{H}^{\boldsymbol{V}^\top}) = \sum_i \lambda_i\theta_i\gamma_i sum(\boldsymbol{u}_i\boldsymbol{u}_i^\top)
\tag{16}
$$

Finally, plug this equation into eq. (11), we can get:

$$\mathcal{L} \le tr(\boldsymbol{H^A H^{V^\top}}) - \frac{1}{N} sum(\boldsymbol{H^A H^{V^\top}})$$

$$= \sum_i \lambda_i \theta_i \gamma_i [1 - \frac{1}{N} sum(\boldsymbol{u_i u_i^\top})]$$

$$\le (1 + N) \sum_i \lambda_i \theta_i \gamma_i \qquad (17a)$$

$$= \frac{1+N}{2} \sum_i \theta_i [\lambda_i^2 + \gamma_i^2 - (\lambda_i - \gamma_i)^2]$$

$$\le \frac{1+N}{2} \sum_i \theta_i [2 - (\lambda_i - \gamma_i)^2]. \qquad (17b)$$

In eq. (17a), we utilize the following lemma:

**Lemma 1.** $sum(\boldsymbol{u_i u_i^\top}) \ge -N^2$.

*Proof.* We have known that $\boldsymbol{u_i^\top u_i} = u_{i1}^2 + \cdots + u_{iN}^2 = 1$, which implies that $\forall j \in [1, N], u_{ij} \in (-1, 1)$. Thus, $\forall j, k \in [1, N], u_{ij} u_{ik} \in (-1, 1)$. Finally, $sum(\boldsymbol{u_i u_i^\top}) = \sum_{j,k} u_{ij} u_{ik} > \sum_{j,k} (-1) = -N^2$. $\qquad\qquad\square$

And in eq. (17b), we utilize the fact that for $\boldsymbol{A}$, amplitudes of its frequencies lay in [-1, 1], while for $\boldsymbol{V}$, amplitudes of its all frequencies equal to 1. Therefore, $\lambda_i^2 \le 1$ and $\gamma_i^2 \le 1$. Similarly for $\mathcal{L}(\boldsymbol{h_i^{V_2}}, \boldsymbol{h_i^{V_1}})$, we also have the same upper bound. Therefore, we have the following results:

$$\mathcal{L}_{InfoNCE} = \sum_i \frac{1}{2} \left( \mathcal{L}(\boldsymbol{h_i^{V_1}}, \boldsymbol{h_i^{V_2}}) + \mathcal{L}(\boldsymbol{h_i^{V_2}}, \boldsymbol{h_i^{V_1}}) \right) \le \frac{1+N}{2} \sum_i \theta_i [2 - (\lambda_i - \gamma_i)^2]. \qquad (18)$$

So, the contrastive invariance is proved. $\qquad\qquad\square$

### A.2 Derivation of optimization objective

In this subsection, we provide detailed derivation of optimization objective in eq. (3). As shown in section 5, we attempt to learn an augmented graph from original graph based on the GAME rule. Based on eigenvalue perturbation formula (2) removing the high-order term $\mathcal{O}(||\Delta \boldsymbol{A}||)$, we have:

$$\Delta \lambda_i \approx \boldsymbol{u_i^\top} \Delta \boldsymbol{A} \boldsymbol{u_i} - \lambda_i \boldsymbol{u_i^\top} \Delta \boldsymbol{D} \boldsymbol{u_i}$$

$$= \sum_{m,n} [(\boldsymbol{u_i u_i^\top}) * \Delta \boldsymbol{A}]_{m,n} - \lambda_i \boldsymbol{u_i^\top} \Delta \boldsymbol{D} \boldsymbol{u_i} \qquad (19a)$$

$$= < \boldsymbol{S_i}, \Delta \boldsymbol{A} > - \lambda_i \boldsymbol{u_i^\top} \Delta \boldsymbol{D} \boldsymbol{u_i}. \qquad (19b)$$

In eq. (19a), we use '*' to represent element-wise product, and use $< \boldsymbol{P}, \boldsymbol{Q} >$ to represent the sum of all elements of $\boldsymbol{P} * \boldsymbol{Q}$ in eq. (19b). Before further derivation, we give a lemma as following:

**Lemma 2.** $|\lambda_i \boldsymbol{u_i^\top} \Delta \boldsymbol{D} \boldsymbol{u_i}| \le N|\lambda_i|$.

This lemma is proved in Appendix A.2.1. Then with Lemma 2, we gather the changes of all eigenvalues together, and have the following two theorems:

**Theorem 5.** *Given* $\Delta_{total} = \sum_i \alpha_i |\Delta \lambda_i|$, *we have*

$$\Delta_{total} \iff \sum_i \alpha_i |< \boldsymbol{S_i}, \Delta \boldsymbol{A} >| \ge \sum_i \alpha_i |< \boldsymbol{S_i}, \Delta_{\boldsymbol{A+}} >| - \sum_j a_j |< \boldsymbol{S_j}, \Delta_{\boldsymbol{A-}} >| = \Delta_+ - \Delta_-,$$

*where* $\alpha_i \ge 0$ *is the weight for the change of* $\lambda_i$. *Formally, we have* $\Delta \boldsymbol{A} = \Delta_{\boldsymbol{A+}} - \Delta_{\boldsymbol{A-}}$, *where* $\Delta_{\boldsymbol{A+}}$ *and* $\Delta_{\boldsymbol{A-}}$ *indicate which edge is added and deleted, respectively.*

**Theorem 6.** *Given N eigenspaces* $[\boldsymbol{S_1}, \ldots, \boldsymbol{S_N}]$, *if* $| < \boldsymbol{S_i}, \Delta \boldsymbol{A} > |$ *is maximized, then* $\forall j$, $| < \boldsymbol{S_j}, \Delta \boldsymbol{A} > | \to 0$.

The proof is given in Appendix A.2.2 and A.2.3. Theorem 6 implies that $\Delta \boldsymbol{A}$ can only be related to one of these eigenspaces. Please recall that we need $|\Delta \lambda_i|$ in $\mathcal{F}_{\mathcal{L}}$ is smaller than that in $\mathcal{F}_{\mathcal{H}}$. Therefore, we first make $\alpha_i$ is larger for $\Delta \lambda_i$ in $\mathcal{F}_{\mathcal{H}}$ than in $\mathcal{F}_{\mathcal{L}}$. Then, we maximize $\Delta_{total}$, and $\Delta \boldsymbol{A}$ will only be related to $\boldsymbol{S}_i$ in $\mathcal{F}_{\mathcal{H}}$. That means $|\Delta \lambda_i|$ in $\mathcal{F}_{\mathcal{H}}$ will be larger. Particularly, Theorem 5 indicates that we can maximize $\Delta_+$ and minimize $\Delta_-$ simultaneously to maximize $\Delta_{total}$. For $\Delta_+$, we can maximize it by following formula:

$$\Delta_+ = \sum_i \alpha_i |<\boldsymbol{S}_i, \Delta_{\boldsymbol{A}+}>| = |\sum_i \alpha_i <\boldsymbol{S}_i, \Delta_{\boldsymbol{A}+}>|$$

$$= |<\sum_i \alpha_i \boldsymbol{u}_i \boldsymbol{u}_i^\top, \Delta_{\boldsymbol{A}+}>| = |<\boldsymbol{U}g(\lambda)\boldsymbol{U}^\top, \Delta_{\boldsymbol{A}+}>| = |<\boldsymbol{C}, \Delta_{\boldsymbol{A}+}>|. \quad (20a)$$

In eq. (20a), we define $\boldsymbol{C} = \boldsymbol{U}g(\lambda)\boldsymbol{U}^\top$ and $\alpha_i = g(\lambda_i)$, where $g(\lambda)$ should be a monotone increasing function. As can be seen, if $\Delta_{\boldsymbol{A}+}$ is learnt to be closed to some predefined $\boldsymbol{C}$, $\Delta_+$ will increase.

**Setting $\boldsymbol{C}$** We notice that the graph spectrum of Laplacian $\boldsymbol{\mathcal{L}}$ does meet our need about $g(\lambda)$. Therefore, we simply set $\boldsymbol{C} = \Theta \boldsymbol{\mathcal{L}}$, where $\Theta$ is a parameter. We change $\Theta$ as the training goes.

With eq. (20a), we give the following optimization objective:

$$\mathcal{J} = <\boldsymbol{C}, \Delta_{\boldsymbol{A}+}>^2 + \epsilon H(\Delta_{\boldsymbol{A}+}) + <\boldsymbol{f}, \Delta_{\boldsymbol{A}+}\mathbb{1}_n - \boldsymbol{a}> + <\boldsymbol{g}, \Delta_{\boldsymbol{A}+}^\top \mathbb{1}_n - \boldsymbol{b}>, \quad (21)$$

where $H(\boldsymbol{P})$ is the entropy regularization, defined as $-\sum_{i,j} \boldsymbol{P}_{i,j}(\log(\boldsymbol{P}_{i,j}) - 1)$ [21], and $\epsilon$ is the weight of this term. This term exposes more edges in $\Delta_{\boldsymbol{A}+}$ to optimization. The last two terms are Lagrange constraint conditions, where $\boldsymbol{f} \in \mathbb{R}^N$ and $\boldsymbol{g} \in \mathbb{R}^N$ are Lagrange multipliers, and $\boldsymbol{a} \in \mathbb{R}^{N \times 1}$ and $\boldsymbol{b} \in \mathbb{R}^{N \times 1}$ are distributions that the row and column sums of $\Delta_{\boldsymbol{A}+}$ should meet.

### A.2.1   Proof of Lemma 2

*Proof.*

$$|\lambda_i \boldsymbol{u}_i^\top \Delta \boldsymbol{D} \boldsymbol{u}_i| = |\lambda_i| |\boldsymbol{u}_i^\top \boldsymbol{diag}(d_1, \ldots, d_N) \boldsymbol{u}_i| = |\lambda_i| \left[ |d_1| \boldsymbol{u}_{i1}^2 + \cdots + |d_N| \boldsymbol{u}_{iN}^2 \right]$$
$$\leq |d|_{max} |\lambda_i| \left[ \boldsymbol{u}_{i1}^2 + \cdots + \boldsymbol{u}_{iN}^2 \right] = |d|_{max} |\lambda_i| \leq N |\lambda_i|.$$

$\square$

In this proof, we utilize two facts: (1) the length of any eigenvector is 1, and (2) the change of degree matrix does not exceed N (in the limiting case, one node becomes from isolated to reaching all other nodes, where $d_{max}$ equals to N).

### A.2.2   Proof of Theorem 5

*Proof.*

$$\Delta_{total} = \sum_i a_i |\Delta \lambda_i|$$

$$= \sum_i a_i \left| <\boldsymbol{S}_i, \Delta \boldsymbol{A}> - \lambda_i \boldsymbol{u}_i^T \Delta \boldsymbol{D} \boldsymbol{u}_i \right|$$

$$\geq \sum_i a_i |<\boldsymbol{S}_i, \Delta \boldsymbol{A}>| - \left| \lambda_i \boldsymbol{u}_i^T \Delta \boldsymbol{D} \boldsymbol{u}_i \right|$$

$$= \sum_i a_i |<\boldsymbol{S}_i, \Delta \boldsymbol{A}>| - \sum_k \left| \lambda_k \boldsymbol{u}_k^T \Delta \boldsymbol{D} \boldsymbol{u}_k \right|$$

$$\geq \sum_i a_i |<\boldsymbol{S}_i, \Delta \boldsymbol{A}>| - N \sum_k |\lambda_k| \quad (22a)$$

$$\geq \sum_i a_i |<\boldsymbol{S}_i, \Delta \boldsymbol{A}>| - N^2 \quad (22b)$$

$$\iff \sum_i a_i |<\boldsymbol{S}_i, \Delta \boldsymbol{A}>| \quad (22c)$$

$$\geq \sum_i a_i |<\boldsymbol{S}_i, \Delta_{\boldsymbol{A}+}>| - \sum_j a_j |<\boldsymbol{S}_i, \Delta_{\boldsymbol{A}-}>|.$$

In eq. (22a), we utilize the result of lemma 2, and in eq. (22b) we again utilize that the amplitudes of frequencies of $\boldsymbol{A}$ are between [-1, 1]. For simplification, we ignore the constant $N^2$ in eq. (22c).  □

### A.2.3  Proof of Theorem 6

*Proof.* To prove the Theorem 6, we firstly prove the following lemma:

**Lemma 3.** $< \boldsymbol{S}_i, \boldsymbol{S}_j >= 0$, when $i \neq j$.

*Proof.* We have that $< \boldsymbol{S}_i, \boldsymbol{S}_j >=< \boldsymbol{u}_i \boldsymbol{u}_i^\top, \boldsymbol{u}_j \boldsymbol{u}_j^\top >= \sum_{m,n} (\boldsymbol{u}_i \boldsymbol{u}_i^\top) * (\boldsymbol{u}_j \boldsymbol{u}_j^\top) = \boldsymbol{u}_i^\top \boldsymbol{u}_j \boldsymbol{u}_j^\top \boldsymbol{u}_i = 0$,

where we take advantage of any two different eigenvectors are orthogonal, or $\boldsymbol{u}_i^\top \boldsymbol{u}_j = 0$.  □

With this lemma, we give the following derivation:

$$2 < \boldsymbol{S}_i, \ \Delta \boldsymbol{A} >= 2\Delta \boldsymbol{A}_{11} \boldsymbol{S}_{i11} + 2\Delta \boldsymbol{A}_{12} \boldsymbol{S}_{i12} + \cdots + 2\Delta \boldsymbol{A}_{NN} \boldsymbol{S}_{iNN}. \tag{23}$$

Then we conduct category discussion on $< \boldsymbol{S}_i, \ \Delta \boldsymbol{A} >$:

1. For $< \boldsymbol{S}_i, \ \Delta \boldsymbol{A} >> 0$, we have:

$$\begin{aligned}
2| < \boldsymbol{S}_i, \ \Delta \boldsymbol{A} > | &= 2\Delta \boldsymbol{A}_{11} \boldsymbol{S}_{i11} + 2\Delta \boldsymbol{A}_{12} \boldsymbol{S}_{i12} + \cdots + 2\Delta \boldsymbol{A}_{NN} \boldsymbol{S}_{iNN} \\
&= \sum_{m,n} \Delta \boldsymbol{A}_{mn}^2 + \boldsymbol{S}_{imn}^2 - (\Delta \boldsymbol{A}_{mn} - \boldsymbol{S}_{imn})^2 \\
&= \sum_{m,n} \Delta \boldsymbol{A}_{mn}^2 + ||\boldsymbol{S}_i||_F^2 - \sum_{m,n} (\Delta \boldsymbol{A}_{mn} - \boldsymbol{S}_{imn})^2 \\
&\leq N^2 + ||\boldsymbol{S}_i||_F^2 - \sum_{m,n} (\Delta \boldsymbol{A}_{mn} - \boldsymbol{S}_{imn})^2.
\end{aligned} \tag{24a}$$

In eq. (24a), we make use of that the change of one element in $\Delta \boldsymbol{A}$ is in $[-1, 1]$. From eq. (24a), we can infer that if $\Delta \boldsymbol{A}$ is learnt to be similar with some eigenspace $\boldsymbol{S}_i$, maximizing the left hand of above equation, then $\sum_{m,n} (\Delta \boldsymbol{A}_{mn} - \boldsymbol{S}_{imn})^2$ will be minimized, which means $\Delta \boldsymbol{A}_{mn} \rightarrow \boldsymbol{S}_{imn}$.

Then, if we test the similarity between $\Delta \boldsymbol{A}$ and another eigenspace $\boldsymbol{S}_j$, we have that:

$$\begin{aligned}
< \boldsymbol{S}_j, \ \Delta \boldsymbol{A} > &= \Delta \boldsymbol{A}_{11} \boldsymbol{S}_{j11} + \Delta \boldsymbol{A}_{12} \boldsymbol{S}_{j12} + \cdots + \Delta \boldsymbol{A}_{NN} \boldsymbol{S}_{jNN} \\
&\Rightarrow \boldsymbol{S}_{i11} \boldsymbol{S}_{j11} + \boldsymbol{S}_{i12} \boldsymbol{S}_{j12} + \cdots + \boldsymbol{S}_{iNN} \boldsymbol{S}_{jNN} \\
&=< \boldsymbol{S}_j, \ \boldsymbol{S}_i >= 0.
\end{aligned}$$

2. For $< \boldsymbol{S}_i, \ \Delta \boldsymbol{A} >< 0$, we have:

$$\begin{aligned}
2| < \boldsymbol{S}_i, \ \Delta \boldsymbol{A} > | &= -2\Delta \boldsymbol{A}_{11} \boldsymbol{S}_{i11} - 2\Delta \boldsymbol{A}_{12} \boldsymbol{S}_{i12} - \cdots - 2\Delta \boldsymbol{A}_{NN} \boldsymbol{S}_{iNN} \\
&= \sum_{m,n} \Delta \boldsymbol{A}_{mn}^2 + \boldsymbol{S}_{imn}^2 - (\Delta \boldsymbol{A}_{mn} + \boldsymbol{S}_{imn})^2 \\
&= \sum_{m,n} \Delta \boldsymbol{A}_{mn}^2 + ||\boldsymbol{S}_i||_F^2 - \sum_{m,n} (\Delta \boldsymbol{A}_{mn} + \boldsymbol{S}_{imn})^2 \\
&\leq N^2 + ||\boldsymbol{S}_i||_F^2 - \sum_{m,n} (\Delta \boldsymbol{A}_{mn} + \boldsymbol{S}_{imn})^2.
\end{aligned}$$

In this case, $\sum_{m,n} (\Delta \boldsymbol{A}_{mn} + \boldsymbol{S}_{imn})^2$ will be minimized, which means $\Delta \boldsymbol{A}_{mn} \rightarrow -\boldsymbol{S}_{imn}$. Then, for another eigenspace $\boldsymbol{S}_j$, we have:

$$\begin{aligned}
< \boldsymbol{S}_j, \ \Delta \boldsymbol{A} > &= \Delta \boldsymbol{A}_{11} \boldsymbol{S}_{j11} + \Delta \boldsymbol{A}_{12} \boldsymbol{S}_{j12} + \cdots + \Delta \boldsymbol{A}_{NN} \boldsymbol{S}_{jNN} \\
&\Rightarrow -\boldsymbol{S}_{i11} \boldsymbol{S}_{j11} - \boldsymbol{S}_{i12} \boldsymbol{S}_{j12} - \cdots - \boldsymbol{S}_{iNN} \boldsymbol{S}_{jNN} \\
&= - < \boldsymbol{S}_j, \ \boldsymbol{S}_i >= 0.
\end{aligned}$$

Above all, we prove that $\Delta \boldsymbol{A}$ can only be related to part of eigenspaces, rather than all of them.  □

## A.3 Proof of Theorem 2

*Proof.* Here, we start from eq. (4b), and replace $(\Delta_{A+})_{ij}$ with $\Delta_{ij}$, shown as following:

$$
\begin{aligned}
f(\Delta_{ij}) = \frac{\partial \mathcal{J}}{\partial(\Delta_{A+})_{ij}} &= m_{ij} + 2\mathcal{C}_{ij}^2 (\Delta_{A+})_{ij} - \epsilon \log(\Delta_{A+})_{ij} + \boldsymbol{f}_i + \boldsymbol{g}_j \\
&= m_{ij} + 2\mathcal{C}_{ij}^2 \Delta_{ij} - \epsilon \log \Delta_{ij} + \boldsymbol{f}_i + \boldsymbol{g}_j,
\end{aligned}
\tag{28}
$$

where the domain of definition is $\Delta_{ij} \in (0,1)$. We observe eq. (28) that $f(0) \to +\infty$ and $f(1) = m_{ij} + 2\mathcal{C}_{ij}^2 + \boldsymbol{f}_i + \boldsymbol{g}_j$. Then, we conduct category discussion.

1. If $f(1) < 0$, we have $\mathcal{C}_{ij}^2 < -\frac{\boldsymbol{f}_i + \boldsymbol{g}_j + m_{ij}}{2}$, and $\boldsymbol{f}_i + \boldsymbol{g}_j + m_{ij} < 0$. In this case, by Existence Theorem of Zero Points, we know that there must be a point making $f(\Delta_{ij}) = 0$, and the loss $\mathcal{J}$ can obtain the extremum. This is the condition (1) in Theorem 2.

2. If $f(1) > 0$, we must let the minimal value of $f(\Delta_{ij}) < 0$, in order to guarantee there exists Zero Points. To get the minimal value, we need take the derivative about $f(\Delta_{ij})$ as follows:

$$
\frac{\partial f(\Delta_{ij})}{\partial \Delta_{ij}} = 2\mathcal{C}_{ij}^2 - \frac{\epsilon}{\Delta_{ij}}.
\tag{29}
$$

From above formula, the minimal value is $\Delta_{ij} = \frac{\epsilon}{2C_{ij}^2}$ by letting $\frac{\partial f(\Delta_{ij})}{\partial \Delta_{ij}} = 0$. Then, we need to meet the following two conditions:

$$
\frac{\epsilon}{2\mathcal{C}_{ij}^2} < 1 \; and \; f(\frac{\epsilon}{2C_{ij}^2}) < 0.
\tag{30}
$$

The solution of above two conditions is the condition (2) in Theorem 2. $\qquad\square$

## A.4 Proof of Theorem 3

*Proof.* To begin with, we introduce some basic definitions, which will be used in this part of proof.

**Definition 1.** *(Hilbert's projective metric)[3] Given $\forall(x, x') \in (\mathbb{R}_+^n)^2$, then the Hilbert's projective metric on $\mathbb{R}_+^n$ is defined as:*

$$
d(x, x') = \log \max_{i,k} \frac{x_i x_k'}{x_i' x_k}.
$$

This is an explicitly defined distance function on a bounded convex subset of the n-dimensional Euclidean space, which can greatly simplified the global convergence analysis of Sinkhorn [7]. This metric satisfies the triangular inequality, and $d(x, x') = 0$ if $\exists s > 0, x = sx'$.

**Definition 2.** *(Contraction ratio)[7] Given a positive $n \times n$ matrix $\boldsymbol{A}$, then the contraction ratio of $\boldsymbol{A}$ is defined as:*

$$
\gamma = \kappa(\boldsymbol{A}) = \sup \left\{ \frac{d(\boldsymbol{A}\boldsymbol{y}, \boldsymbol{A}\boldsymbol{y}')}{d(\boldsymbol{y}, \boldsymbol{y}')} : \; \boldsymbol{y}, \boldsymbol{y}' \in \mathbb{R}_+^n, \; \boldsymbol{y} \neq \alpha\boldsymbol{y}' \right\}
$$

Specially, if we set $\aleph = \sup \left\{ d(\boldsymbol{A}\boldsymbol{y}, \boldsymbol{A}\boldsymbol{y}') : \; \boldsymbol{y}, \boldsymbol{y}' \in \mathbb{R}_+^n, \; \boldsymbol{y} \neq \alpha\boldsymbol{y}' \right\}$, then $\gamma = \frac{\aleph^{1/2} - 1}{\aleph^{1/2} + 1}$.

With these two definitions, the authors in [22] give the following lemma:

**Lemma 4.** *One has*

$$
|| \log(\boldsymbol{P}^{(l)}) - \log(\boldsymbol{P}^\star) ||_\infty \leq \frac{1}{1 - \gamma} \{ d(\boldsymbol{r}^{(l)}, \boldsymbol{a}) + d(\boldsymbol{c}^{(l)}, \boldsymbol{b}) \},
\tag{31}
$$

*where $\boldsymbol{P}^{(l)}$ and $\boldsymbol{P}^\star$ are the intermediate result after lth Sinkhorn's Iteration and the final result, respectively. $\boldsymbol{r}^{(l)}$ and $\boldsymbol{c}^{(l)}$ are the row and column sum vectors of $\boldsymbol{P}^{(l)}$, and $\boldsymbol{a}$ and $\boldsymbol{b}$ are two predefined distributions.*

Now, please recall that we utilize the trick to operate Sinkhorn's Iteration in eq. (6). In this case, we set $l$ in eq. (31) as 0, so the left hand of equation will be changed as:

$$
|| \log(\boldsymbol{P}^{(0)}) - \log(\boldsymbol{P}^\star) ||_\infty = || \log(\boldsymbol{K}_+) - \log(\Delta_{\boldsymbol{A}+}) ||_\infty.
\tag{32}
$$

This is because before iteration, $\boldsymbol{P}^{(0)}$ is actually $\boldsymbol{K}_+$, and $\Delta_{\boldsymbol{A}+}$ is the final result of iteration. Then, we have the following derivation:

$$
\begin{aligned}
|\log((\Delta_{\boldsymbol{A}+})_{ij}) - \log((\boldsymbol{K}_+)_{ij})| &\leq ||\log(\Delta_{\boldsymbol{A}+}) - \log(\boldsymbol{K}_+)||_\infty \\
&\leq \frac{1}{1-\gamma}\{d(\boldsymbol{r}^{(0)}, \boldsymbol{a}) + d(\boldsymbol{c}^{(0)}, \boldsymbol{b})\} \\
\left|\epsilon\log((\Delta_{\boldsymbol{A}+})_{ij}) - 2 < \boldsymbol{C}, \Delta'_{\boldsymbol{A}+} > \boldsymbol{C}_{ij}\right| &\leq \frac{1}{\epsilon(1-\gamma)}\{d(\boldsymbol{r}^{(0)}, \boldsymbol{a}) + d(\boldsymbol{c}^{(0)}, \boldsymbol{b})\} \\
\left|\epsilon\log((\Delta_{\boldsymbol{A}+})_{ij}) - 2\boldsymbol{C}_{ij}^2(\Delta'_{\boldsymbol{A}+})_{ij} - m_{ij}\right| &\leq \frac{1}{\epsilon(1-\gamma)}\{d(\boldsymbol{r}^{(0)}, \boldsymbol{a}) + d(\boldsymbol{c}^{(0)}, \boldsymbol{b})\} \\
\left|\epsilon((\Delta_{\boldsymbol{A}+})_{ij} - 1) - 2\boldsymbol{C}_{ij}^2(\Delta'_{\boldsymbol{A}+})_{ij} - m_{ij}\right| &\leq \frac{1}{\epsilon(1-\gamma)}\{d(\boldsymbol{r}^{(0)}, \boldsymbol{a}) + d(\boldsymbol{c}^{(0)}, \boldsymbol{b})\} \\
\because \left|\epsilon(\Delta_{\boldsymbol{A}+})_{ij} - 2\boldsymbol{C}_{ij}^2(\Delta'_{\boldsymbol{A}+})_{ij}\right| - \epsilon - |m_{ij}| &\leq \left|\epsilon((\Delta_{\boldsymbol{A}+})_{ij} - 1) - 2\boldsymbol{C}_{ij}^2(\Delta'_{\boldsymbol{A}+})_{ij} - m_{ij}\right| \\
\therefore \left|\epsilon(\Delta_{\boldsymbol{A}+})_{ij} - 2\boldsymbol{C}_{ij}^2(\Delta'_{\boldsymbol{A}+})_{ij}\right| &\leq \frac{1}{\epsilon(1-\gamma)}\{d(\boldsymbol{r}^{(0)}, \boldsymbol{a}) + d(\boldsymbol{c}^{(0)}, \boldsymbol{b})\} + \epsilon + |m_{ij}| \\
\left|\frac{\epsilon}{2\boldsymbol{C}_{ij}^2}(\Delta_{\boldsymbol{A}+})_{ij} - (\Delta'_{\boldsymbol{A}+})_{ij}\right| &\leq \frac{1}{2\boldsymbol{C}_{ij}^2\epsilon(1-\gamma)}\{d(\boldsymbol{r}^{(0)}, \boldsymbol{a}) + d(\boldsymbol{c}^{(0)}, \boldsymbol{b})\} \\
&\quad + \frac{\epsilon + |m_{ij}|}{2\boldsymbol{C}_{ij}^2}.
\end{aligned}
\tag{33}
$$

If we set $\alpha = \frac{\epsilon}{2\boldsymbol{C}_{ij}^2}$, we have:

$$
\left|\alpha(\Delta_{\boldsymbol{A}+})_{ij} - (\Delta_{\boldsymbol{A}+})'_{ij}\right| \leq \frac{\alpha}{\epsilon^2(1-\gamma)}\{d(r^{(0)}, \boldsymbol{a}) + d(c^{(0)}, \boldsymbol{b})\} + \alpha(1 + \frac{|m_{ij}|}{\epsilon}). \tag{34}
$$

$\square$

## B  More details of Section 3

### B.1  Experimental settings

In Fig. 2, we show the GCL model used in following case study. Here, we provide more experimental settings about case study. As description, we use a shared GCN to encode $\boldsymbol{A}$ and $\boldsymbol{V}$ and get their nodes embeddings as $\boldsymbol{H_A}$ and $\boldsymbol{H_V}$. In the training, we set dimensions of $\boldsymbol{H_A}$ and $\boldsymbol{H_V}$ as 8, learning rate as 0.001, $\tau$ in eq. 1 as 0.5 and weight decay as 0. We train the model 300 epochs, and then test the quality of $\boldsymbol{H_A}$. In the test phrase, we follow DGI, and the split of each dataset is consistent with that in table 4 in Appendix D.2, and we only test the case where training set contains 20 nodes per class.

### B.2  More results of case study

In section 3, we design a simple GCL framework to contrast $\boldsymbol{A}$ and generated $\boldsymbol{V}$. When constructing $\boldsymbol{V}$, we conduct augmentations in $\mathcal{F_L}$ (or $\mathcal{F_H}$) by adding eigenspaces in $\mathcal{F_L}$ (or $\mathcal{F_H}$) back in low-to-high frequency order. Here, we consider the inverse high-to-low frequency order. The process is shown in Fig. 11, where $\boldsymbol{V}$ augmenting 20% in $\mathcal{F_L}$ is $\boldsymbol{u}_{0.8*N/2}\boldsymbol{u}_{0.8*N/2}^\top + \cdots + \boldsymbol{u}_{(N-1)/2}\boldsymbol{u}_{(N-1)/2}^\top + \boldsymbol{u}_{N/2}\boldsymbol{u}_{N/2}^\top + \cdots + \boldsymbol{u}_N\boldsymbol{u}_N^\top$. Similarly, $\boldsymbol{V}$ augmenting 20% in $\mathcal{F_H}$ is $\boldsymbol{u}_1\boldsymbol{u}_1^\top + \cdots + \boldsymbol{u}_{N/2}\boldsymbol{u}_{N/2}^\top + \boldsymbol{u}_{1.8*N/2}\boldsymbol{u}_{1.8*N/2}^\top + \cdots + \boldsymbol{u}_N\boldsymbol{u}_N^\top$. We take Cora as an example, and report the results in Fig. 12. As can be observed, we get the similar conclusions as in section 3: (1) From the curve marked by "Low", the performances remain stably low until the lowest-frequency components are involved; (2) From the curve marked by "High", the performances continually rise, as more high-frequency components are involved. Therefore, based on observations, we emphasize the importance of lowest-frequency and more high-frequency information in GCL. So, whether we

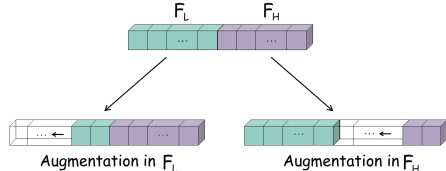

Figure 11: New generation of V, where we follow high-to-low frequency order in both low-frequency and high-frequency part.

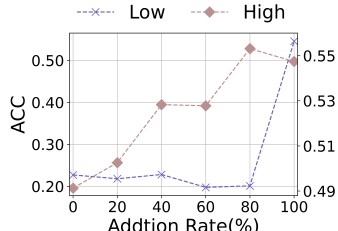

Figure 12: The results of inverse adding order on Cora.

incrementally add in the low-to-high or high-to-low frequency order, as long as lowest-frequency and more high-frequency information is preserved in $V$, the results will be better.

## C   Another Experimental Analysis of GAME Rule

We substitute $A^2$ (two-hop of $A$) for augmentation $V$ in the case study. In Fig. 13, we simultaneously plot the graph spectra of $A$ and $A^2$. It can be seen that $A$ and $A^2$ meet GAME rule. Next, we compare the performances of this pair with two other pairs: $A$ and $A$, $A^2$ and $A^2$, where the spectra of two augmentations are the same and not meet GAME rule in each pair. We randomly run 10 times for each pairs, and report the average accuracy on four datasets. The results are shown in Table 3, where we can see that the pair of $A$ and $A^2$ actually excels the other two pairs.

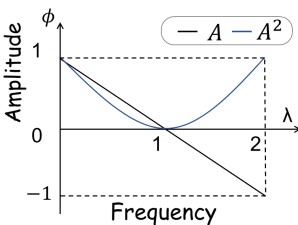

Figure 13: The spectrum of $A$ and $A^2$.

Table 3: Performance of there different pairs of augmentations to verify the GAME rule.

| Pair of Augmentations | Cora | Citeseer | BlogCatalog | Flickr |
|:---:|:---:|:---:|:---:|:---:|
| $A$ & $A$ | 37.0±6.1 | 35.4±3.9 | 50.6±3.2 | 26.6±2.6 |
| $A$ & $A^2$ | **53.7±3.2** | **44.7±5.0** | **63.1±4.6** | **33.7±2.3** |
| $A^2$ & $A^2$ | 33.3±2.1 | 35.8±4.1 | 56.2±2.1 | 28.2±1.6 |

## D   Experimental Details

### D.1   Implementation details

For two semi-supervised GNN models (GCN, GAT), we utilize the original codes from their authors, and train the models in an end-to-end way. For six graph contrastive learning methods, we obey the traditional GSL setting: we first train each model by optimizing corresponding objective, and get the embeddings for all nodes; then, we feed the node embeddings into a logistic regression to evaluate the quality of embeddings. We use the source codes provided by authors, and follow the settings in their original papers with carefully tune.

For the proposed SpCo, we search on combination coefficient $\eta$ from 0.1 to 1.2 with step 0.1. For $\epsilon$, we test it ranging from {1e-1, 1e-2, 1e-3, 1e-4}. We tune the iteration $Iter$ for Sinkhorn's Iteration in algorithm 1 from {1, 2, 3}. Finally, we carefully select total epochs $T$ and update epochs $\Omega$ for target model, and tune the scope $\mathbb{S}$ from one-hop or two-hop neighbors.

For fair comparisons, we randomly run 10 times and report the average results for all methods. For the reproducibility, we report the related parameters in Appendix D.3.

### D.2 Datasets and baselines

We choose the five commonly used Cora, Citeseer, Pubmed [11], BlogCatalog and Flickr [16] for evaluation. The first three datasets are citation networks, where nodes represent papers, edges are the citation relationship between papers, node features are comprised of bag-of-words vector of the papers and labels represent the fields of papers. For them, we choose 500 nodes for validation, 1000 nodes for test. The BlogCatalog dataset is a social network with bloggers and their social relationships from the BlogCatalog website. Node features are constructed by the keywords of user profiles, and the labels represent the topic categories provided by the authors. The Flickr dataset is an image and video hosting website, where users interact with each other via photo sharing. It is a social network where nodes represent users and edges represent their relationships. For these two methods, We choose 1000 nodes for validation, 1000 nodes for test. We uniformly select three label rates for the training set (i.e., 5, 10, 20 labeled nodes per class) for all datasets. Please notice that we adopt classical splits [11] on the first three datasets for the 20 labeled nodes per class setting. The details of these datasets are summarized in table 4.

Table 4: The statistics of the datasets.

| Dataset | Nodes | Edges | Classes | Features | Training | Validation | Test |
|---------|-------|-------|---------|----------|----------|------------|------|
| Cora | 2708 | 10556 | 7 | 1433 | 35/70/140 | 500 | 1000 |
| Citeseer | 3327 | 9228 | 6 | 3703 | 30/60/120 | 500 | 1000 |
| BlogCatalog | 5196 | 343486 | 6 | 8189 | 30/60/120 | 1000 | 1000 |
| Flickr | 7575 | 479476 | 9 | 12047 | 45/90/180 | 1000 | 1000 |
| Pubmed | 19717 | 88651 | 3 | 500 | 15/30/60 | 500 | 1000 |

These five datasets used in experiments can be found in these URLs:

- Cora, Citeseer, Pubmed: `https://github.com/tkipf/gcn`
- BlogCatalog, Flickr: `https://github.com/mengzaiqiao/CAN`

And the publicly available implementations of Baselines can be found at the following URLs:

- GCN: `https://github.com/tkipf/gcn`
- GAT: `https://github.com/PetarV-/GAT`
- DGI: `https://github.com/PetarV-/DGI`
- MVGRL: `https://github.com/kavehhassani/mvgrl`
- GRACE: `https://github.com/CRIPAC-DIG/GRACE`
- GCA: `https://github.com/CRIPAC-DIG/GCA`
- GraphCL: `https://github.com/Shen-Lab/GraphCL`
- CCA-SSG: `https://github.com/hengruizhang98/CCA-SSG`

### D.3 Hyper-parameters settings

We implement SpCo in PyTorch, and list some important parameter values in our model in table 5.

During experiments, we notice that Pubmed is a large graph, which will consume much time to operate Sinkhorn's Iteration. Meanwhile, we find that our SpCo is model-agnostic, which means that the Sinkhorn's Iteration is independent of some specific model. Therefore, we choose to calculate the iteration results for Pubmed and store them for utilization. Specifically, we firstly calculate 10 intermediate iteration results for Pubmed, and then for each method, we take $num$ intermediate results at equal intervals, and feed them one by one into the target model every $epoch\_inter$ epochs. Therefore, the relative parameter values are shown in table 6.

### D.4 More experiment results

#### D.4.1 Node classification

In this section, we provide more experimental results on node classification, where we choose 5 and 10 labeled nodes per class as training set respectively, and keep the same validation and test set. The

Table 5: The values of parameters used in SpCo on Cora, Citeseer, BlogCatalog and Flickr.

| Method | Dataset | $T$ / patience for early stopping | $\Omega$ | $\mathbb{S}$(hop) | $\eta$ | $\epsilon$ | $Iter$ |
|---|---|---|---|---|---|---|---|
| DGI+SpCo | Cora | 40 (patience) | 20 | 1 | 0.1 | 1.0 | 3 |
| | Citeseer | 30 (patience) | 20 | 1 | 0.5 | 1.0 | 3 |
| | BlogCatalog | 30 (patience) | 30 | 2 | 0.3 | 1e-2 | 3 |
| | Flickr | 30 (patience) | 30 | 1 | 0.3 | 1e-2 | 2 |
| GRACE+SpCo | Cora | 300 ($T$) | 30 | 1 | 0.5 | 1.0 | 3 |
| | Citeseer | 150 ($T$) | 20 | 1 | 1.0 | 1e-2 | 3 |
| | BlogCatalog | 800 ($T$) | 300 | 1 | 1.0 | 1e-2 | 3 |
| | Flickr | 1300 ($T$) | 300 | 1 | 1.0 | 1e-1 | 2 |
| CCA-SSG+SpCo | Cora | 40 ($T$) | 15 | 1 | 0.5 | 1e-2 | 3 |
| | Citeseer | 15 ($T$) | 5 | 1 | 0.1 | 1e-1 | 3 |
| | BlogCatalog | 75 ($T$) | 10 | 1 | 0.1 | 1e-1 | 3 |
| | Flickr | 100 ($T$) | 30 | 1 | 0.5 | 1e-2 | 2 |

Table 6: The values of parameters used in SpCo on Pubmed.

| Method | $T$ / patience for early stopping | $num$ | $epoch\_inter$ | $\mathbb{S}$(hop) | $\eta$ | $\epsilon$ | $Iter$ |
|---|---|---|---|---|---|---|---|
| DGI+SpCo | 30 (patience) | 3 | 100 | 1 | 0.1 | 1e-2 | 1 |
| GRACE+SpCo | 1500 ($T$) | 7 | 100 | 1 | 1.0 | 1e-2 | 1 |
| CCA-SSG+SpCo | 170 ($T$) | 7 | 15 | 1 | 0.1 | 1e-2 | 1 |

relative results are given in Table 7. Again, we can see that additional SpCo generally improves the performance of corresponding baselines on all datasets, which comprehensively demonstrates that SpCo can boost node classification in an effective way. Especially, we notice that the loss used in CCA-SSG is originated from canonical correlation analysis, which is definitely different from InfoNCE loss (1). But, we can still boost the results of CCA-SSG by plugging SpCo, indicating the necessity to design effective augmentations.

### D.4.2 Visualisation of graph spectrum

In this section, we mainly supplement the visualisation of graph spectrum on Cora, shown in Fig. 14. Again, $A\_$ and $A$ have a smaller difference in $\mathcal{F}_{\mathcal{L}}$ than in $\mathcal{F}_{\mathcal{H}}$, indicating they satisfy the GAME rule. And $V_1$ and $V_2$ got from them also present the same pattern. Therefore, with contrastive invariance theorem 1, they will instruct encoder to capture more low-frequency information, and hence improve the results.

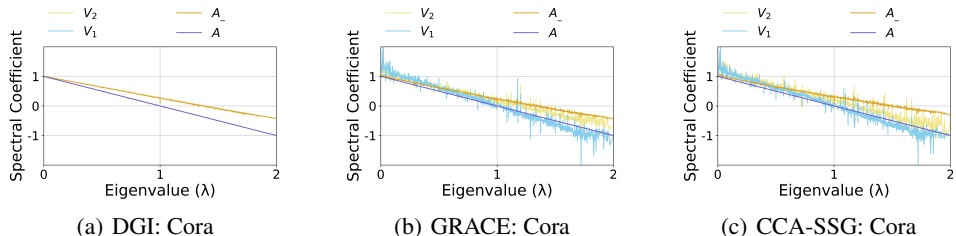

    (a) DGI: Cora              (b) GRACE: Cora          (c) CCA-SSG: Cora

Figure 14: The visualisation of graph spectrum on Cora.

### D.4.3 Hyper-parameter sensitivity

In this section, we show more results about $\epsilon$ on Cora, shown in Fig. 15. Then, we give more analyses about hyper-parameter $\eta$, and test its sensitivity on Cora and BlogCatalog.

**Analysis of $\eta$.** The $\eta$ in eq. (8) controls the strength that $\Delta_{\boldsymbol{A}}$ changes the original $\boldsymbol{A}$. If $\eta$ is larger, $\Delta_{\boldsymbol{A}}$ will give more impact on $\boldsymbol{A}$, and the margin between $\boldsymbol{A}\_$ and $\boldsymbol{A}$ is larger in $\mathcal{F}_{\mathcal{H}}$. We again range the value of $\eta$ from 0.2 to 1.2, and the results are shown in Fig 16. In the figure, the shallower color is, the better performance is. And the maximal value is marked as red star. We can see that

Table 7: Quantitative results ($\%\pm\sigma$) on node classification. The better results compared with target model are in bold.

| Datasets | Metrics | Splits | GCN | GAT | DGI | DGI+SpCo | MVGRL | GRACE | GRACE+SpCo | GCA | GraphCL | CCA-SSG | CCA+SpCo |
|---|---|---|---|---|---|---|---|---|---|---|---|---|---|
| Cora | Ma-F1 | 5 | 68.4±1.4 | 74.4±0.3 | 76.1±0.4 | **76.2±0.3** | 76.2±0.5 | 70.5±2.0 | **71.4±2.1** | 70.7±2.8 | 75.1±0.4 | 73.4±0.7 | **73.7±0.6** |
| | | 10 | 72.9±0.5 | 75.7±0.5 | 77.8±0.7 | **78.2±0.4** | 78.4±0.8 | 73.9±1.5 | **75.7±1.6** | 75.5±1.4 | 77.8±0.7 | 77.1±0.4 | **78.4±1.0** |
| | Mi-F1 | 5 | 69.6±1.5 | 75.8±0.4 | 77.3±0.5 | **77.4±0.4** | 77.0±0.7 | 70.7±1.8 | **71.5±2.1** | 71.5±2.8 | 76.4±0.3 | 73.6±0.7 | **73.8±0.8** |
| | | 10 | 73.5±0.6 | 76.5±0.5 | 79.1±0.6 | **79.3±0.5** | 79.4±0.8 | 74.9±1.6 | **76.5±1.3** | 76.4±1.7 | 79.0±0.8 | 77.4±0.5 | **78.7±1.5** |
| Citeseer | Ma-F1 | 5 | 48.2±1.5 | 55.3±0.4 | 62.6±1.2 | **63.3±0.6** | 58.3±0.9 | 58.1±1.5 | **59.2±1.3** | 52.3±2.5 | 62.1±1.1 | 57.2±1.4 | **59.0±2.1** |
| | | 10 | 63.8±0.7 | 64.6±0.5 | 64.9±0.7 | **65.0±0.5** | 62.8±0.8 | 61.9±0.9 | **62.1±0.5** | 59.3±1.4 | 64.4±0.6 | 63.6±1.1 | **65.2±0.5** |
| | Mi-F1 | 5 | 53.4±1.1 | 61.8±1.9 | 67.4±1.5 | **68.0±0.6** | 63.0±1.1 | 62.5±1.5 | **63.9±1.6** | 54.6±3.0 | 66.8±1.3 | 61.4±1.5 | **62.5±2.1** |
| | | 10 | 66.8±0.8 | 67.9±0.6 | 68.9±0.5 | **69.2±0.4** | 67.2±0.8 | 65.2±0.8 | **66.0±1.0** | 61.8±1.7 | 68.5±0.7 | 67.5±1.1 | **69.7±0.9** |
| BlogCatalog | Ma-F1 | 5 | 61.4±2.8 | 55.7±3.9 | 60.2±1.8 | **65.9±1.3** | 69.0±7.1 | 56.6±1.5 | **58.4±0.6** | 62.5±0.8 | 55.6±3.1 | 67.9±0.4 | **68.2±0.5** |
| | | 10 | 68.2±1.8 | 65.0±1.3 | 67.1±1.1 | **70.5±0.4** | 76.2±4.0 | 64.9±0.8 | **65.8±0.9** | 69.9±0.4 | 63.3±1.6 | **67.6±0.5** | 67.5±0.3 |
| | Mi-F1 | 5 | 63.3±2.7 | 58.1±3.1 | 61.8±1.7 | **67.2±1.1** | 69.9±7.1 | 58.9±1.5 | **60.4±0.6** | 64.4±0.7 | 57.2±3.3 | 69.0±0.4 | **69.2±0.4** |
| | | 10 | 69.5±1.8 | 65.9±1.5 | 67.7±1.2 | **71.3±0.5** | 76.7±3.9 | 65.8±0.8 | **66.8±1.0** | 70.8±0.5 | 63.9±1.7 | 68.3±0.5 | **68.4±0.3** |
| Flickr | Ma-F1 | 5 | 32.4±1.6 | 27.9±1.3 | 20.9±1.3 | **24.3±1.0** | 21.7±3.0 | 21.0±5.5 | **31.3±1.0** | 34.6±0.4 | 22.3±1.3 | 30.6±1.1 | **31.4±1.0** |
| | | 10 | 42.8±1.1 | 32.8±2.3 | 26.2±2.3 | **26.8±0.6** | 29.1±3.8 | 32.6±1.7 | **33.5±4.1** | 39.3±0.6 | 25.9±2.2 | 32.3±0.6 | **33.9±0.2** |
| | Mi-F1 | 5 | 35.4±1.0 | 30.9±0.2 | 21.9±1.0 | **26.6±1.1** | 23.0±3.0 | 23.6±4.2 | **32.7±0.9** | 35.7±0.4 | 23.4±1.1 | 33.4±0.8 | **33.8±0.1** |
| | | 10 | 44.2±1.0 | 35.3±1.4 | 27.1±2.0 | **29.8±0.6** | 30.5±3.8 | 34.9±1.0 | **35.9±2.4** | 40.4±0.5 | 27.3±1.9 | 35.2±0.9 | **36.2±0.3** |
| PubMed | Ma-F1 | 5 | 69.3±0.7 | 70.1±0.4 | 72.3±1.0 | **73.0±0.4** | 73.8±0.7 | **75.5±0.7** | 75.4±0.6 | 70.9±1.5 | 70.7±0.4 | 72.1±0.8 | **73.1±0.4** |
| | | 10 | 73.1±1.1 | 72.5±0.3 | 73.2±0.7 | **74.2±0.5** | 72.9±0.5 | 73.2±1.6 | **73.4±1.4** | 75.0±0.1 | 72.3±0.4 | 73.0±1.0 | **74.2±0.6** |
| | Mi-F1 | 5 | 68.4±0.8 | 69.8±0.4 | 72.2±1.2 | **73.0±0.6** | 73.2±1.2 | 75.5±0.7 | **76.0±0.6** | 70.5±1.8 | 70.4±0.6 | 71.5±0.8 | **72.7±0.4** |
| | | 10 | 72.7±1.3 | 72.2±0.4 | 72.5±0.9 | **73.3±0.6** | 71.9±0.6 | 72.1±1.9 | **72.9±1.6** | 74.6±0.2 | 71.3±0.5 | 71.9±1.2 | **73.3±0.7** |

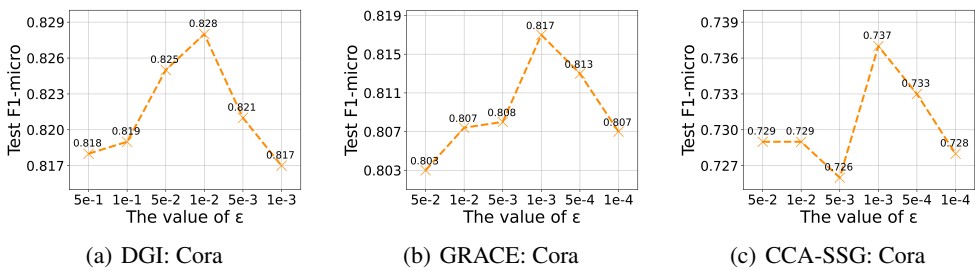

(a) DGI: Cora    (b) GRACE: Cora    (c) CCA-SSG: Cora

Figure 15: Analysis of the hyper-parameter $\epsilon$ on Cora.

the performances are relative stable in this range. Please note that, this range is reasonable. If the lower $\eta$ is tested, our SpCo will have a smaller influence. And if the higher $\eta$ is tested, our SpCo will dominate the combination in eq. (8), which will destroy the original $\boldsymbol{A}$ heavily.

## D.5 Operating environment

The environment where our code runs is shown as follows:

- Operating system: Linux version 3.10.0-693.el7.x86_64
- CPU information: Intel(R) Xeon(R) Silver 4210 CPU @ 2.20GHz
- GPU information: GeForce RTX 3090

## D.6 Detailed descriptions of target models

As introduced in experiments, we plug our SpCo into three existing GCL frameworks, including DGI [28], GRACE [35] and CCA-SSG [33], to roundly verify the effectiveness of SpCo. Therefore, it is necessary to briefly introduce their mechanisms in this section.

- **DGI**: This method deploys contrastive learning between local patch and summary vector. Specifically, the authors utilize GCN to encode the original graph and get all node embeddings, which is viewed as local patch. Then, they design a readout function to summarize

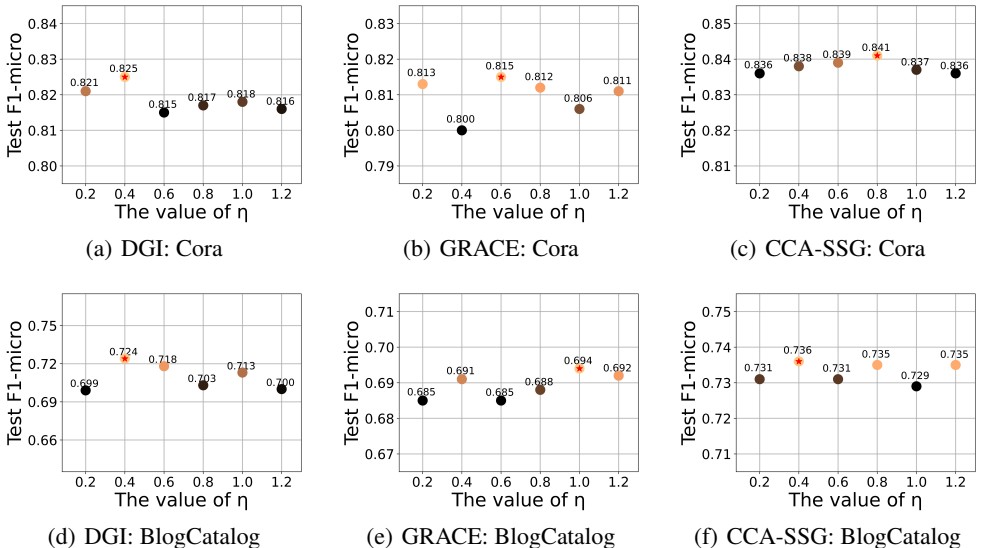

Figure 16: Analysis of the hyper-parameter $\eta$ on Cora and BlogCatalog.

node embeddings into one vector, which represents the global information of the graph. In optimization, DGI views every node in the graph as the positive sample of the summary vector, while views nodes in corrupted graph as negative samples. Different from InfoNCE loss, DGI uses BCE loss as the objective.

- **GRACE**: This method uses feature mask and random edge dropping as two augmentation strategies from feature and topology levels. Specifically, the authors perform two strategies at the same time but with different ratios to generate two augmentations, respectively. Then, they obey traditional InfoNCE loss, where the positive sample of one node is just it own embedding in the other augmentation, and other nodes are viewed as negative samples.

- **CCA-SSG**: This method inherits previous augmentation strategies, like feature mask and random edge dropping. However, the main contribution of this work is to involve canonical correlation analysis into GCL framework and propose a new objective, which maximizes the correlation between two views and prevent degenerated solutions simultaneously.

# E  Related Work

**Graph Neural Networks.** Recently, Graph Neural Networks (GNNs) have attracted considerable attentions, which can be broadly divided into two categories, spectral-based and spatial-based. Spectral-based GNNs are inheritance of graph signal processing, and define graph convolution operation in spectral domain. For example, [2] utilizes Fourier bias to decompose graph signals; [5] employs the Chebyshev expansion of the graph Laplacian to improve the efficiency. For another line, spatial-based GNNs greatly simplify above convolution by only focusing on neighbors. For example, GCN [11] simply averages information of one-hop neighbors. GraphSAGE [8] only randomly fuses a part of neighbors with various poolings. GAT [27] assigns different weights to different neighbors. More detailed surveys can be found in [29].

**Graph Contrastive Learning.** Graph Contrastive Learning has shown its distinguished capacity in unsupervised setting, and many studies have been proposed. In this paper, we mainly focus on the various augmentation strategies. Specifically, DGI [28] contrasts between local node embedding and global summary vector. MVGRL [9] proposes several strategies based on diffusion or distance matrix. For {GRACE [35], GCA [36], GraphCL [32], CCA-SSG [33]}, they can roughly be gathered into the same category: the random edge and node perturbation. There are some frameworks aiming to learn an adaptive augmentation with the help of different principles, such as AD-GCL [26], InfoGCL [30], DGCL [13] and GASSL [31]. But these studies mainly focus on graph classification task. Besides, we also notice that some papers propose augmentation free [12].