# OpenReview forum: "Revisiting Graph Contrastive Learning from the Perspective of Graph Spectrum"
_NeurIPS.cc/2022/Conference — NeurIPS 2022 Accept_

### Official Review · Reviewer_g6CU · 2022-07-10

**Rating:** 8
**Confidence:** 5
**Soundness:** 4 excellent
**Presentation:** 3 good
**Contribution:** 4 excellent

**Summary:**

This paper mainly focuses on a fundamental problem in graph contrastive learning (GCL) domain, which is dedicated to design a general augmentation rule to uplift the performance of current GCL frameworks. The authors first commence from a complete case study to figure out a general augmentation rule, GAME rule, with a contrastive invariance theorem. Then, to practice this rule, the authors design the SpCo architecture, a general plug-in by plentiful and rigorous derivation. They also conduct adequate experiments to demonstrate the effectiveness of SpCo and analyze some relative characters.

**Questions:**

1. As mentioned above, the authors should give more descriptions about theorem 3.

2. The figure 1(a) draws a skeleton about general framework of GCL, but I notice some components are neglected, like projection head and feature mask.

3. This paper mainly starts from InfoNCE loss, while BCE loss used by DGI is another main branch in GCL. So, what will the result should be for BCE loss?

4. This paper argues that the difference between low-frequency parts between two views should be smaller than that of high-frequency parts to reduce the negative impact of high-frequency parts. If so, can we directly filter out the high-frequency parts and only keep the low-frequency parts to achieve the same goal?

 5. I notice that the authors utilize Sinhorn’s Iteration, which involves some matrix multiplication operations. I worry about the computational complexity.

**Limitations:**

Yes, the authors have addressed the limitations and potential negative societal impact of their work.

**Strengths And Weaknesses:**

Strengths:
1. Motivation: As claimed in the paper, there lacks a deeper insight concerning about a general rule behind various augmentations. This is a core and valuable problem for GCL, and this paper has made a beneficial step towards this target.

2. Methodology: This paper proposes the GAME rule and follows it to devise SpCo framework. Each part is supported by adequate theorems and derivations, which contributes to this technically solid paper.

3. Experiments: The authors employ extensive experiments, where the base models see various degrees of improvement.

4. Contexture: This paper is well-organized, and the literature is coherent and readable. Meanwhile, the illustrations are esthetic and concise enough.

Weakness:
1. As pointed out in conclusion, this paper seems only effective to homophily graphs, but failed to heterophily graphs. Anyway, this flaw is not so severe, since homophily is a basic assumption in unsupervised settings. I suggest the authors to further promote this paper to adapt to each case flexibility.

2. The proposed contrastive invariance theorem seems a little limited. For example, the authors restrain two augmentations as A and V, rather than two random augmentations. It may be a higher requirement.

3. Some details are not explained so clearly. For instance, followed by theorem3, there are some descriptions about its function, which is a little unclear.

---

> ### Author Response · Authors · 2022-08-02
> **Response (2/2)**
>
> 5. > I notice that the authors utilize Sinhorn’s Iteration, which involves some matrix multiplication operations. I worry about the computational complexity.
>
> **Response**: We think that with the additional SpCo, the computational complexity of other existing GCL methods will not increase. In formula 10, we will make $\Delta_{\boldsymbol{A}}$ sparse by multiplying $\mathbb{S}$, which guarantees that $\boldsymbol{A}$ will be added by a sparse matrix. Also, the computation of $\Delta_{\boldsymbol{A}}$ is implemented on CPU, so this process will not account for any computational resource of GPU.

---

> > ### Comment · Reviewer_g6CU · 2022-08-08
> > **Reply to author’s response**
> >
> > Dear authors,
> >
> > Thank you for the detailed response. It has addressed my concerns. I would like to keep my score to support this paper.

---

> ### Author Response · Authors · 2022-08-02
> **Response (1/2)**
>
> We sincerely thank the Reviewer for all the comments and it is a great honor for us for your enjoying our paper. We have addressed all your questions below and hope they have clarified all confusion you had about our work.
>
> 1. > 1. As mentioned above, the authors should give more descriptions about theorem 3.
>
> **Response**: Theorem 3 states that $\Delta \boldsymbol{A}$ will be only related to one eigenspace $\boldsymbol{S_i}$, while orthogonal with other eigenspaces. With the maximizatoin of $\Delta_{total}$, we set $\alpha_i$ is larger for $\Delta \lambda_i$ in $\mathcal{F_H}$ than in $\mathcal{F_L}$, which will induce $\Delta \boldsymbol{A}$ to be related to $\boldsymbol{S_i}$ in $\mathcal{F_H}$. In this case, $\Delta \lambda_i$ in $\mathcal{F_H}$ will be larger, which achieves our target that high-frequency part should present a large difference.
>
> 2. > The figure 1(a) draws a skeleton about general framework of GCL, but I notice some components are neglected, like projection head and feature mask.
>
> **Response**: Yes, your opinion is right. We actually neglect projection head and feature mask in figure 1(a). (1) For projection head, it is a common part in classical GCL framework. However, in this paper, we do not aim to discuss the effect of different network architectures. In contrast, we mainly focus on the study of general rule of topological augmentation. So, we keep the projection head as other methods when conducting experiments, but do not mentioned in figure 1(a). (2) For feature mask, it is one mainstream augmentation designed for features in graph. However, we tempt to revisit the current graph contrastive learning from graph spectral theory, which has a strong connection with graph topology. Specifically, our goal in this paper is to build the relations among different topology augmentations, and point out a general rule uncovering which two contrasted graphs are better for GCL from graph spectral view. Moreover it is well known that feature masking is actually a general augmentation strategy, not only designed for graph contrastive learning. Thus we mainly focus on analyzing topology augmentation in this paper, as clearly demonstrated in many different parts in this paper, e.g., Line 288. So, we just keep the topological augmentations in figure 1(a), and neglect feature mask.
>
> 3. > This paper mainly starts from InfoNCE loss, while BCE loss used by DGI is another main branch in GCL. So, what will the result should be for BCE loss?
>
> **Response**: Actually, there are three categories losses used in the baselines: BCE loss (DGI, MVGRL), InfoNCE loss (GRACE, GCA, GraphCL) and CCA loss (CCA-SSG). And our theorem starts from InfoNCE loss, which is a classical contrastive loss widely used both in graph and in CV or NLP. However, as mentioned in paper, our main motivation is to answer what information should we preserve in two contrasted augmentations, and figure out GAME rule to indicate which conditions two graphs should satisfy. This rule is independent on some certain selection of contrastive loss. In fact, we have tested our SpCo on BCE loss in experments, for example the DGI+SpCo performs better than pure DGI. We will focus on the further theoretical analysis on BCE loss in the future.
>
> 4. > This paper argues that the difference between low-frequency parts between two views should be smaller than that of high-frequency parts to reduce the negative impact of high-frequency parts. If so, can we directly filter out the high-frequency parts and only keep the low-frequency parts to achieve the same goal?
>
> **Response**: The results will be bad if we directly filter out the high-frequency parts and only keep the low-frequency parts. Let us take the curve marked by “High” In figure 3 (a)(b)(c)(d) as example. The start point is marked as $\boldsymbol{V}_0$, where addition rate equals to 0. In this case,  $\boldsymbol{V}_0$ contains all low-frequency information without any high-frequency information. We also illustrate the graph spectrum of  $\boldsymbol{V}_0$ at an anonymous URL https://imgur.com/a/zURYCRL, because of the limitation of inserting figures in this textbox. You can see that $\boldsymbol{V}_0$ only keep the low-frequency part in generated  $\boldsymbol{V}$ as you said. However in figure 3, the performance of $\boldsymbol{V}_0$ is almost the worst. So, this example shows that we should enhance the difference of high-frequency components between two  contrasted graphs, rather than directly filter our high-frequency information.

---

### Official Review · Reviewer_gfze · 2022-07-13

**Rating:** 5
**Confidence:** 3
**Soundness:** 3 good
**Presentation:** 3 good
**Contribution:** 3 good

**Summary:**

This paper discusses the reason that some contrastive learning methods work well in the view of graph spectrum. Based on their survey, the authors proposed the general graph augmentation rule (GAME rule) which defines what kind of pair of augmentations is optimal contrastive pair. Finally, they proposed a plug-in method for graph augmentation based on this rule. The experiments and theoretical analysis show the effectiveness of the proposed approach.

**Questions:**

1. In Table 2, why not report the experiments of MVGRL/GCA/GaphCL + SpCo?
2. In line 151, “With Eq. (2), we calculate the eigenvalues on Cora after each augmentation”, where the eigenvalues on Cora could be directly obtained, how Eq. (2) is used here?
3. In line 167, it was mentioned that “In other words, the invariance of two augmentations is enhanced from the point of view of graph spectrum.” The details were limited, and more explanations should be provided.
4. It seems that the proposed SpCo focuses on the high-frequency information, but actually, GNNs often face over smoothing because of the low-frequency information. So, can this method eliminate the over smoothing problem?

Some typos:
- In line 178, "we use ‘*’ to" -> "we use `*’ to".
-  In lines 179, 194, "eq." -> "Eq.".
- Some notations are not consistent, for example $\bf{u}$_i in Eq. (3a).

**Limitations:**

Yes

**Strengths And Weaknesses:**

Strengths：
- This paper is well structured. The authors first propose problems, then analyze them with extensive experiments, and finally resolve these issues through the proposed method and theorem, which makes this paper can be well understood.
- The analysis of graph contrastive learning with a graph spectrum view is interesting, which provides a new aspect for GCL.
- The experiments and theories are sufficient to demonstrate the effectiveness.

Weaknesses：
- Although the paper is well structured, it is difficult to understand some theorems and formulas. The notations used in the paper is too complex, more explanations and working principles of the proposed method should be given.
- From Table 2, the performance improvement of SpCo is very limited comparing to the other methods by taking into account the corresponding standard deviation values.

---

> ### Author Response · Authors · 2022-08-02
> **Response (2/2)**
>
> 4. > It seems that the proposed SpCo focuses on the high-frequency information, but actually, GNNs often face over smoothing because of the low-frequency information. So, can this method eliminate the over smoothing problem?
>
> **Response**: **(1)** Please note that SpCo focuses on the **differences** of amplitudes of frequencies between **two** given contrasted graphs, rather than only high-frequency or low-frequency information in a graph. In fact, we emphasize the importance of the **lowest-frequency** information and the **high-frequency** information, stated in Line 43-44. Essentially, SpCo points out a way to change the original matrix into a new one, based on GAME rule, and form the optimal contrastive pair for GCL, which emphasizes the consistency of lowest-frequency information and diversity of high-frequency information between two contrasted graphs.
>
>  **(2)** As mentioned in (1), we mainly study the differences of amplitudes of frequencies between two given contrasted graphs, rather than only high-frequency or low-frequency information.  Therefore, there is no obvious connection between our work and over-smoothing.
>
> 5. > Some typos errors
>
> **Response**: Thanks, we will correct these mistakes in the future version.
>
> You have mentioned that there are some difficult theorems and formulas, and complex notations. We are sorry to make you confusion when reading our paper. If you have any questions about our technique details, we are looking forward to discussing with you. We will polish our writing furthermore.

---

> > ### Comment · Reviewer_gfze · 2022-08-08
> > **No further concerns**
> >
> > Thanks for the authors' response. Based on the new experimental results, it can be seen that there are marginal gains of SpCo with other models. Therefore I will keep my initial rating.

---

> ### Author Response · Authors · 2022-08-02
> **Response (1/2)**
>
> We sincerely thank the Reviewer for valuable opinions and concerns about our work. It is our obligation to describe more details and give more explanations, including experiments, motivation and writing. We really hope that these further efforts can alleviate your confusion.
>
> 1. > In Table 2, why not report the experiments of MVGRL/GCA/GaphCL + SpCo?
>
> **Response**: Here, we choose the base models by taking contrastive losses into account. Baselines in Table 2 can be roughly divided into three categories based on their contrastive losses: BCE loss (DGI, MVGRL), InfoNCE loss (GRACE, GCA, GraphCL) and CCA loss (CCA-SSG). Our aim is to evaluate the performance of our model and validate the generalization in different categories, so we select one baseline from each category to integrate with SpCo. Besides, to answer your question, we also test SpCo with MVGRL, GCA, and GraphCL on Cora, where the results are shown in the following table. We can see that SpCo outperforms base models again. This further implies the effectiveness of our model, and we will open the source code.
>
> |       | split |  MVGRL   |     MVGRL+SpCo      |   GCA    |      GCA+SpCo       | GraphCL  |    GraphCL+SpCo     |
> | :---: | :---: | :------: | :-----------------: | :------: | :-----------------: | :------: | :-----------------: |
> | Ma-F1 |   5   | 76.2±0.5 | $\textbf{76.7±0.5}$ | 70.7±2.8 | $\textbf{72.7±1.8}$ | 75.1±0.4 | $\textbf{75.5±0.8}$ |
> | Ma-F1 |  10   | 78.4±0.8 | $\textbf{78.7±0.3}$ | 75.5±1.4 | $\textbf{76.7±1.4}$ | 77.8±0.7 | $\textbf{78.3±0.6}$ |
> | Ma-F1 |  20   | 81.5±0.5 | $\textbf{82.2±0.6}$ | 79.9±1.1 | $\textbf{80.2±0.5}$ | 80.7±0.9 | $\textbf{81.8±0.7}$ |
> | Mi-F1 |   5   | 77.0±0.7 | $\textbf{77.5±0.6}$ | 71.5±2.8 | $\textbf{73.4±1.7}$ | 76.4±0.3 | $\textbf{77.0±0.7}$ |
> | Mi-F1 |  10   | 79.4±0.8 | $\textbf{79.7±0.4}$ | 76.4±1.7 | $\textbf{77.8±1.2}$ | 79.0±0.8 | $\textbf{79.4±0.6}$ |
> | Mi-F1 |  20   | 82.8±0.4 | $\textbf{83.4±0.6}$ | 81.1±1.0 | $\textbf{81.6±0.6}$ | 82.3±0.9 | $\textbf{83.0±0.7}$ |
>
> 2. > In line 151, “With Eq. (2), we calculate the eigenvalues on Cora after each augmentation”, where the eigenvalues on Cora could be directly obtained, how Eq. (2) is used here?
>
> **Response**: Thanks for your comments. Actually, given two graphs (the original matrix $\boldsymbol{A}$ and the augmented matrix $\boldsymbol{A'}$), the eigenvalues $\lambda'$ of $\boldsymbol{A'}$ can be directly obtained by eigenvalue decomposition. However, the obtained eigenvalues $\lambda'$' are **unordered** compared with previous eigenvalues $\lambda$ of $\boldsymbol{A}$. That is to say, for certain $\lambda_i$ of $\boldsymbol{A}$, we cannot figure out which eigenvalue of $\boldsymbol{A'}$ matches to it after decomposition, so we cannot calculate the change $\Delta \lambda$ for i-th frequency component $\lambda_i$. In contrast, Eq. (2) clearly points out the relationship of eigenvalues between and after augmentation. Therefore, with Eq. (2), we can calculate $\Delta \lambda$ and then plot this change in figure 5.
>
> 3. > In line 167, it was mentioned that “In other words, the invariance of two augmentations is enhanced from the point of view of graph spectrum.” The details were limited, and more explanations should be provided.
>
> **Response**: Firstly, we set the amplitudes of $i$th frequency component of two contrasted graphs as $\lambda_i$ and $\gamma_i$. If $\lambda_i\approx\gamma_i$, these two contrasted graphs show the **invariance** at $i$th frequency. Next, with the training going on, $\mathcal{L}_{total}$ will be maximized according to InfoNCE loss in formula (1), which leads to the maximization of the right hand of Theorem 1. In this case, if at $i$th frequency, $(\lambda_i-\gamma_i)^2$ is smaller (invariance at $i$th frequency is more obvious), then $[2-(\lambda_i-\gamma_i)^2]$ will be larger. So, larger $\theta_i$ should be learned to be assigned to larger  $[2-(\lambda_i-\gamma_i)^2]$ term, so that the right hand of Theorem 1 will be maximized. So, we state that Theorem 1 indicates that with contrastive learning, the encoder can capture the invariance between two contrasted graphs from spectrum domain.

---

### Official Review · Reviewer_3xAc · 2022-07-15

**Rating:** 5
**Confidence:** 2
**Soundness:** 3 good
**Presentation:** 1 poor
**Contribution:** 3 good

**Summary:**

The paper notes prior work on Graph Contrastive Learning (GCL), then proposes a new concept based on which to produce the contrasting graph: by focusing on changing the spectrum of the graph. In particular, the authors assert the "GAME rule," which roughly states that a good contrastive pair of graphs differs more in high-frequency components than in low-frequency ones. They then run an experiment on real-world graphs to support this assertion, followed by proof of a theorem upper bounding the GCL loss, which also suggests application of the GAME rule. The authors then theoretically derive their SpCo algorithm, which allows for producing contrasting graphs in a manner that obeys GAME, and perform experiments to show that it (1) can improve prior methods' performance and (2) indeed produces graphs which tend to obey GAME.

**Questions:**

Based on the Figure 1(b) and Figure 6, the modified $\mathbf{A_-}$ looks somewhat like a low-pass filtered version of $\mathbf{A}$. Have the authors confirmed that setting $\mathbf{A_-}$ to just be a low-pass filtered version of $\mathbf{A}$ will not work as well?

I found the description of the experiment in Section 3 difficult to understand. Is there perhaps a mistake in Lines 125-126 or Figure 3? I don't see how the addition rate can go up to 80%; based on the description, it seems like it should only go up to 50% at most. Also, why are the eigenvalues ignored and set to 1? Finally, it seems a bit strange that both high and low frequency components are added incrementally in the low-to-high frequency order.
It also seems like this section in general should go after the experimental setup, e.g., hyperparameters and how link prediction is done, has been fully described.

I have similar issues with the other experimental sections. It would be helpful to clearly describe the motivation, the full setup, and how the results support the conclusion.

The statement of Theorem 1 could be more rigorous: For what network model is the loss minimized? (This is stated, with fairly strong assumptions, but only in the appendix.) And what is meant by 'limited value'?

The theory work in Section 5.1 is quite dense, and could use more discussion of the intuition behind the steps. I recommend moving most of it to the appendix, just keeping a sketch in the main work, and moving Algorithm 2 into the main work, making the algorithm description more self-contained (i.e., naming all parameters, including $\Theta$ and $\epsilon$, and describing how Eq. 8 and 10 are implemented), and providing more discussion of the intuition of the algorithm.

Minor comments:
+ Line 85: Typo on equation for unnormalized Laplacian (should be degree, not identity matrix)
+ Line 94: Typo "weaken" -> "weakened"
+ Line 131: "the performance achieves the best" -> "the best performance is achieved"
+ Line 136: "specturm" typo
+ Line 142: "with both of" typo
+ Line 176: "tempt" -> "attempt" typo?
+ It wasn't clear to me what is meant by "samples" (Line 102) in the explanation of GCL, especially since the example loss in Equation 1 does not seem stochastic.

**Limitations:**

The authors note the limitation that the work is only applicable to homophilous graphs, not heterophilous ones, which is a useful remark. They could additionally note the assumptions required for the main theory result.
There is not much discussion of potential negative societal impacts beyond that the statement this work could improve GCL methods in the future; there is some possibility, if desired, to discuss potential negative societal impacts of improved GCL methods.

**Strengths And Weaknesses:**

Strengths
+ The paper attempts to advance understanding and application of a recently-popular type of method, GCL.
+ The core idea, at least as described, of using contrastive pairs which mostly preserve low-frequency information while allowing high-frequency information to change seems reasonable and interesting.
+ The paper seems to ground its ideas in theory.

Weaknesses
+ Descriptions of some ideas, procedures, and experimental setups are unclear.
+ Descriptions are also rather dense at times, leaving me with a limited idea of the motivation and intuition.

Notes
+ It is difficult for me to evaluate the soundness of the paper given the presentation issues discussed below.

---

> ### Author Response · Authors · 2022-08-02
> **Response (4/4)**
>
> 7. > The statement of Theorem 1 could be more rigorous: For what network model is the loss minimized? (This is stated, with fairly strong assumptions, but only in the appendix.) And what is meant by 'limited value'?
>
> **Response**: Thanks for your suggestion. The Theorem 1 just describes the optimization of loss of traditional GCL, whose backbone encoder is GCN, which has claimed in Line 100. So, the GCN is used for the loss minimized. We will revise theorem 1 by adding this point.
>
> The 'limited value' means that $|\theta_i|<M$, for $i\in[1, N]$, where $M$ is some upper bound of the size of $\theta$. That is because $\theta_i$ is related to the parameters $\boldsymbol{W}$ of GCN, which is shown in Appendix A.1. Meanwhile, it is a common technique to perform L2 norm on $\boldsymbol{W}$ to restrain the value of $\boldsymbol{W}$ in the training, so the value of $\theta_i$ is also limited correspondingly.
>
> 8. > The theory work in Section 5.1 is quite dense, and could use more discussion of the intuition behind the steps. I recommend moving most of it to the appendix, just keeping a sketch in the main work, and moving Algorithm 2 into the main work, making the algorithm description more self-contained (i.e., naming all parameters, including Θ and ϵ, and describing how Eq. 8 and 10 are implemented), and providing more discussion of the intuition of the algorithm.
>
> **Response**: Thanks for your useful advices. Actually we have already moved all the proofs to the appendix, because here we aim to show the complete process of the derivation of algorithm, this may cause the dense content. However, according to your suggestion, we will carefully revise this part to make it more clear and understandable. Next, we will introduce the mainline of our algorithm.
>
> The whole derivation in Section 5.1 aims to generate a topological augmentation $\boldsymbol{A}$_ from original $\boldsymbol{A}$, with the guidance of GAME rule. We first start from eigenvalue perturbation formula (2) to derivate the difference of the amplitude of $i$th frequency $\Delta \lambda_i$ between $\boldsymbol{A}$_ and $\boldsymbol{A}$. According to GAME rule, $\boldsymbol{A}$_ and $\boldsymbol{A}$ should have a smaller margin in low-frequency part, so we need to require $|\Delta \lambda_i|<|\Delta \lambda_j|$ when $i<j$. To achieve this target, we prove that we should maximize the right hand of formula (4a), exactly $|<\boldsymbol{\mathcal{C}}, \Delta_{A+}>|$, where $\boldsymbol{\mathcal{C}}=\boldsymbol{U}g(\lambda)\boldsymbol{U}^\top$,  $g(\lambda)$ should be a monotone increasing function about $\lambda$, and $\Delta_{A+}$ indicates which edge is added for $\boldsymbol{A}$_ , compared with $\boldsymbol{A}$. To learn a better $\Delta_{A+}$, besides maximizing $|<\boldsymbol{\mathcal{C}}, \Delta_{A+}>|$, we then consider other two requirements about $\Delta_{A+}$: one is that more edges in $\Delta_{A+}$ should be trained (entropy regularization), the other is that $\Delta_{A+}$ should be restrained by row sum and column sum distribution. This leads to the final objective in formula (6). From formula (7) to formula (9), we discuss how to solve the objective with the help of Sinkhorn's Iteration, and give the solution of $\Delta_{A+}$ in formula (8) and solution of $\Delta_{A-}$ in Line 227. So, with the calculated $\Delta_{A+}$ and $\Delta_{A-}$, we can obtain the final $\Delta_{A}=\Delta_{A+}-\Delta_{A-}$, which shows how to implement GAME rule and the expected difference $\Delta_{A}$ between $\boldsymbol{A}$_ and original $\boldsymbol{A}$.
>
> 9. > Typo errors
>
> **Response**: Thanks for pointing out these typo errors. We will update them.
>
> 10. > It wasn't clear to me what is meant by "samples" (Line 102) in the explanation of GCL, especially since the example loss in Equation 1 does not seem stochastic.
>
> **Response**: "samples" in Line 102 does not mean the results of randomly sampling. Rather, "positive samples" and "negative samples" are two concepts in GCL. For one target node, "positive samples" means the nodes that should have embeddings close to the target node, while  "negative samples" means the nodes that should have embeddings being away from the target node. In general, "positive samples" and  "negative samples" are determined aforehand, and fixed during the training. In equation 1, for target node $i$, we view itself in $V_2$ as its positive sample in $V_1$, and all other nodes in $V_2$ as its negative samples in $V_1$.

---

> > ### Comment · Reviewer_3xAc · 2022-08-08
> > **Response to Authors' Comments**
> >
> > Thank you for the thorough response.
> > My main concern is still the clarity of presentation. While the descriptions here help clarify the experiments and intuition for the theory, I think there is significant room for improvement in the writing of the paper itself. However, I am generally persuaded by the authors' arguments here, and the contribution seems significant, so I have raised my score.
> >
> > Two small parting notes:
> > + "The Theorem 1 just describes the optimization of loss of traditional GCL, whose backbone encoder is GCN, which has claimed in Line 100. So, the GCN is used for the loss minimized. We will revise theorem 1 by adding this point."
> > One would assume that the GCN model is used, but the proof in the appendix further assumes that there is no non-linearity, which is not typical in practice, and makes the assumption rather strong. This is why I thought the full assumptions must be clearly stated in the main body, where the theorem is stated.
> >
> > + If at all possible, I think it would greatly increase the readability of the paper if Algorithm 2 were added to the main paper and written in a more self-contained way as I described above.

---

> > > ### Author Response · Authors · 2022-08-09
> > > **Thanks for your support**
> > >
> > > We are very delighted to address some concerns by our further explanations. Meanwhile, we really appreciate your advices to improve our paper: (1) No non-linearity assumption. As you said, it is necessary to state this point in the theorem, which will be supplemented in the revision. (2) Adding algorithm 2 in the main paper. That's actually an important suggestion to make our paper more self-contained. We will add this algorithm in the main paper in the future version. Additionally, we will also polish our presentation and add more details shown in this rebuttal to clarify some key insights of our paper.

---

> ### Author Response · Authors · 2022-08-02
> **Response (3/4)**
>
> 6. > I have similar issues with the other experimental sections. It would be helpful to clearly describe the motivation, the full setup, and how the results support the conclusion.
>
> **Response**: Thanks for your suggestion. Here, we give further explanations for experiments.
>
> **(1)** For the experiments in section 4,
>
> **Motivation:** We aim to verify the correctness of GAME rule that whether two contrasted graphs satisfying GAME rule can perform better in downstream tasks.
>
> **The full setup:** We conduct node classification task on Cora, where the split is the same with case study mentioned in question 5. We keep the same pipeline with case study in section 3 <https://imgur.com/a/IfFyqdq>, where we only replace the inputs with two other kinds of contrasted graphs: $\boldsymbol{A}$ and existing augmentations, or {$\boldsymbol{A}$ and $\boldsymbol{A}$, $\boldsymbol{A}$ and $\boldsymbol{A^2}$, $\boldsymbol{A^2}$ and $\boldsymbol{A^2}$}.
>
> **How the results support the conclusion:**
>
> **a.**  $\boldsymbol{A}$ and existing augmentations proposed by other work. In figure 5, we show the nine mainstream topological augmentations with their graph spectra. Next, let us take the PPR matrix as the example. Compared with the spectrum of $\boldsymbol{A}$ (shown in the left bottom in the figure), we can see that when $\lambda\in[0, 1]$, the amplitude of frequency in PPR is close to that in $\boldsymbol{A}$. But, when $\lambda\in(1, 2]$, the amplitude of frequency in PPR is always greater than 0, but that in $\boldsymbol{A}$ is always lower than 0, which shows a clear disparity. Therefore, $\boldsymbol{A}$ and PPR perfectly meet the need of GAME rule, and they are "optimal contrastive pair". And actually, the performance contrasting between $\boldsymbol{A}$ and PPR is also the best in table 1.
>
> **b.** We test the performances when contrasting {$\boldsymbol{A}$ and $\boldsymbol{A}$, $\boldsymbol{A}$ and $\boldsymbol{A^2}$, $\boldsymbol{A^2}$ and $\boldsymbol{A^2}$}, where the results are reported in table 3 in Appendix C. In figure 10 in Appendix C, we show the graph spectra of $\boldsymbol{A}$ and $\boldsymbol{A^2}$, where we can see that the difference of amplitude in low-frequency part is smaller than that in high-frequency part. Therefore, $\boldsymbol{A}$ and $\boldsymbol{A^2}$ are a pair of "optimal contrastive pair". Actually, contrasting between $\boldsymbol{A}$ and $\boldsymbol{A^2}$ performs better than contrasting between $\boldsymbol{A}$ and $\boldsymbol{A}$ or $\boldsymbol{A^2}$ and $\boldsymbol{A^2}$, as shown in table 3.
>
> In conclusion, we prove the effectiveness of GAME rule in above two cases.
>
> **(2)** For the experiments in section 6,
>
> **Motivation:** we conduct our main experiments to test if our proposed SpCo can improve the performance of base models. If so, we test if these gains are brought by contrasting between 'optimal contrastive pair'.
>
> **The full setup:** The main experiment is node classification, where we integrate our SpCo with three existing GCL models (DGI, GRACE, CCA-SSG). We report the detailed experimental setup in section 6 and Appendix D.1.
>
> **How the results support the conclusion:** Firstly, as shown in table 2, our Spco can generally improve the performances of the base models (DGI, GRACE, CCA-SSG). Furthermore, we visualize the spectra of $\boldsymbol{A}$ and $\boldsymbol{A}$_ generated by our SpCo in figure 7. It can be seen that compared with $\boldsymbol{A}$, $\boldsymbol{A}$_ keeps the lowest-frequency components while ascends the high-frequency components, which is consistent with the target of SpCo shown in figure 6 that we want to enlarge the margin in high-frequency part between $\boldsymbol{A}$ and $\boldsymbol{A}$_ . In this case, $\boldsymbol{A}$ and $\boldsymbol{A}$_ meet the GAME rule and this is the reason why SpCo can uplift base models. In section 6.3, we test some important hyperparameters of SpCo, and give more detailed analyses of the model characters.

---

> ### Author Response · Authors · 2022-08-02
> **Response (2/4)**
>
> 3. > Also, why are the eigenvalues ignored and set to 1?
>
> **Response**: In the Section 3, we aim to explore the effect of different spectral information contained in eigenspaces $u_iu_i^\top$ on GCL. Usually, the eigenvalues $\lambda_i$ acts as a prior that whether the $u_iu_i^\top$ of $i$th frequency is strengthened or weakened. For example, if $\lambda_i=0$, the $u_iu_i^\top$ of $i$th frequency will be totally neglected. Meanwhile, if $\lambda_i$ is relatively large, the effect of $u_iu_i^\top$ will be overemphasized. So, if we keep the original eigenvalues, which are different from each other in general case, we cannot distinguish whether the effect on GCL is brought by different eigenspaces ($u_iu_i^\top$) or by prior $\lambda_i$. Therefore, to test the effect of different $u_iu_i^\top$, we should set $\lambda_i=1$ to avoid the influence from eigenvalues. On the other hand, we follow [1] here, where in figure 3, the authors use graph Fourier basis ($u_i$ in our paper) to filter noisy features without $\lambda_i$, who aim to verify low-frequency information in original feature is more important for machine learning task.
>
> [1] Nt H, Maehara T. Revisiting graph neural networks: All we have is low-pass filters. arXiv:1905.09550, 2019.
>
> 4. > Finally, it seems a bit strange that both high and low frequency components are added incrementally in the low-to-high frequency order.
>
> **Response**: In section 3, our motivation is to explore which part of frequency components should have a smaller margin and which part should have a larger one between two contrasted graphs from graph spectrum domain. Specifically, we fix one of two contrasted views as adjacency matrix $\boldsymbol{A}$, and the other is generated $\boldsymbol{V}$. By adding different rates of eigenspaces in $\mathcal{F_L}$ and $\mathcal{F_H}$, we find that if we keep the **lowest-frequency** and **more high-frequency** components in generated $\boldsymbol{V}$, the performance will be better. In this case, $\boldsymbol{A}$ and $\boldsymbol{V}$ present a small margin in $\mathcal{F_L}$ and a large difference in $\mathcal{F_H}$ according to figure 4, which contributes to our GAME rule where two contrasted graphs are required to follow a larger difference in $\mathcal{F_H}$ and smaller difference in $\mathcal{F_L}$.
>
> From previous descriptions, we emphasize the importance of **lowest-frequency** and **more high-frequency** information in GCL. So, whether we incrementally add in the low-to-high or high-to-low frequency order, as long as lowest-frequency and more high-frequency information is preserved in $\boldsymbol{V}$, the results will be better. To further validate this point, we conduct the case study on Cora where we obey **high-to-low** frequency order to augment in both $\mathcal{F_L}$ and $\mathcal{F_H}$, and results are shown in figure 1 in URL <https://imgur.com/a/bi0Wrvr>, where we also illustrate the process of addition in figure 2.
>
> $\textbf{(1)}$ Augmentation in $\mathcal{F_L}$. From the curve marked by “Low” in figure 1, it is obvious that the performances remain stably low until the lowest-frequency components are involved.
>
> $\textbf{(2)}$ Augmentation in $\mathcal{F_H}$. From the curve marked by “High” in figure 1, we can also see that the performances continually rise, as more high-frequency components are involved.
>
> In conclusion, whether addition in high-to-low frequency order or low-to-high frequency order, the observations will be consistent which is shown in Section 3.
>
> 5. > It also seems like this section in general should go after the experimental setup, e.g., hyperparameters and how link prediction is done, has been fully described.
>
> **Response**: Thanks for your suggestion. Here, we do not report the hyperparameters of model used in case study because of the limited pages. Here, we provide more details of the model. The diagram of the model is given in figure 1 in URL <https://imgur.com/a/IfFyqdq>.
>
> Specifically, the input is the adjacency matrix $\boldsymbol{A}$ and generated $\boldsymbol{V}$. Then, these two contrasted graphs are encoded by a shared one-layer GCN, and obtain their embeddings $\boldsymbol{H_A}$ and $\boldsymbol{H_V}$. We use the InfoNCE loss in formula 1 in Line 108 to train the whole model. In the training, we set dimensions of $\boldsymbol{H_A}$ and $\boldsymbol{H_V}$ as 8, learning rate as 0.001, $\tau$ in formula 1 as 0.5 and weight decay as 0. We train the model 300 epochs, and then test $\boldsymbol{H_A}$. In the test phrase, we follow DGI, and the split of each dataset is consistent with that in table 4 in Appendix D.2, and we only test the case where training set contains 20 nodes per class. According to your suggestion, we will add above details in the appendix.

---

> ### Author Response · Authors · 2022-08-02
> **Response (1/4)**
>
> We sincerely thank the Reviewer for spending time and providing valuable feedback. We appreciate all of your so detailed suggestions and we have addressed all your questions below by providing our responses as well as our additional experimental results. We really look forward to assisting to have a better understanding of our paper.
>
> 1. > Based on the Figure 1(b) and Figure 6, the modified $\boldsymbol{A}$_ looks somewhat like a low-pass filtered version of $\boldsymbol{A}$. Have the authors confirmed that setting $\boldsymbol{A}$_ to just be a low-pass filtered version of $\boldsymbol{A}$ will not work as well?
>
> **Response**: Thanks for your comment. Actually, Figure 1(b) and Figure 6 are just used to illustrate our GAME rule, i.e., give two graph augmentations $\boldsymbol{A}$_ and $\boldsymbol{A}$, the difference of amplitudes of high frequencies in these two graphs should be larger than that of low frequencies. So from our GAME rule, we can see that what we emphasize is the **DIFFERENCE** of **two** graphs $\boldsymbol{A}$_ and $\boldsymbol{A}$, rather than what the frequency is in only one graph. Based on this, $\boldsymbol{A}$_ is not necessarily a low-pass filtered version of A. Here we can understand this point from the following examples:
>
> (1) In Line 157-158, we test the GAME rule using three cases, where one is we contrast between $\boldsymbol{A}$ and $\boldsymbol{A}^2$ and we get the best results in table 3. While, as shown in Figure 10, $\boldsymbol{A}^2$ is actually not a low-pass filtered version of $\boldsymbol{A}$, because the amplitudes are both 1 when  $\lambda=0$ and $\lambda=2$. Meanwhile, the high-frequency parts of $\boldsymbol{A}$ and $\boldsymbol{A}^2$ show a larger difference.
>
> (2) Let us check the curve marked by “High” In figure 3 (a)(b)(c)(d). We use $\boldsymbol{V}_0$ to represent a graph with no high frequency information (i.e., addition rate = 0). In this case, $\boldsymbol{V}_0$ contains all low-frequency information. Because we cannot insert figures into our response here, for better visual presentation, we plot the spectrum of $\boldsymbol{V}_0$ at an anonymous URL https://imgur.com/a/zURYCRL). In this figure, $\forall \lambda\in[0,1]$, the amplitudes all equal to 1, while  $\forall \lambda\in[1,2]$, the amplitudes all equal to 0. We can see that despite $\boldsymbol{V}_0$ is a low-pass filtered version of A, the performance is almost the worst, as shown in figure 3.
>
> 2. > I found the description of the experiment in Section 3 difficult to understand. Is there perhaps a mistake in Lines 125-126 or Figure 3? I don't see how the addition rate can go up to 80%; based on the description, it seems like it should only go up to 50% at most.
>
> **Response**: Thanks for your comment and your understanding is right, and we are sorry for this mistake in Lines 125-126. Let us take augmentation 80% in $\mathcal{F_L}$ for example. In this case, we keep the whole $\mathcal{F_H}$, and only retain 80% of $\mathcal{F_L}$ from low-to-high frequency order $\left(\boldsymbol{u_1u_1}^\top+\dots+\boldsymbol{u_{0.8*{N/2}}u_{0.8*{N/2}}}^\top+\boldsymbol{u_{N/2}u_{N/2}}^\top+\dots+\boldsymbol{u_Nu_N}^\top\right )$. We will revise this typo.

---

### Official Review · Reviewer_ndDo · 2022-07-16

**Rating:** 3
**Confidence:** 5
**Soundness:** 2 fair
**Presentation:** 2 fair
**Contribution:** 1 poor

**Summary:**

In this paper, the authors explore the topological augmentation of GCL from the spectral domain. Based on a series of theoretical analysis, they propose a contrastive invariance theorem and discover a general augmentation (GAME) rule. Then, they propose a general augmentation plug-in based on the GAME rule, i.e., SpCo, to boost existing GCL methods. Extensive experiments verify the effectiveness of SpCo.

**Questions:**

For the questions and suggestions, please refer to W1—W5.


**Limitations:**

Yes.

**Strengths And Weaknesses:**

Pros:

S1: The problem of graph contrastive learning is important and interesting.

S2: The proposed method has a theoretical basis.

Cons:

W1: The notations in this paper is confusing. For example, in Line 80 and Line 81, $\xi$ -> $\mathcal{E}$, N -> $\textit{N}$, $\textbf{A}$ -> $\textbf{\textit{A}}$.

W2: The proposed method has a serious limitation. In the paper, the authors fix one of the two views as the raw adjacency matrix, but do not discuss the two augmented views; in addition, some other augmentation operations (such as feature masking) are not discussed.

W3: Incomplete experiments. Apart from node classification, graph classification is one of the important tasks in graph contrastive learning. However, in this paper, the authors only show the results on node classification and ignore graph classification. In addition, in Table 2, why do not use MVGRL+SpCo (considering that MVGRL seems to be the best baseline used in this paper)?

W4: No proof of 'optimal contrastive pair'. In Figure 1 and other parts, the authors say that 'optimal contrastive pair'. How to prove its optimality? If not, please replace 'optimal' with other words.

W5: Unclear experimental description. For example, In Table 1, the authors do not state what dataset is used in the experiment; in Figure 3, what is the meaning of 'ACC'?

---

> ### Author Response · Authors · 2022-08-02
> **Response (3/3)**
>
> 4. > No proof of 'optimal contrastive pair'. In Figure 1 and other parts, the authors say that 'optimal contrastive pair'. How to prove its optimality? If not, please replace 'optimal' with other words.
>
> **Response**: Thanks for your comment. Please note that 'optimal contrastive pair' is a definition rather than a theorem or some lemmas, so actually we cannot prove a definition. This definition describes a pair of contrasted graphs satisfying GAME rule. Specifically, 'optimal contrastive pair' requires the difference of low-frequency parts between the pair of graphs is smaller that that of high-frequency parts. And we have verified the GAME rule from both experiments and theorem. The usage of 'optimal' just indicates the effectiveness of this rule.
>
> On the other hand, our proposed SpCo aims to learn a new augmentation $\boldsymbol{A}$_ from $\boldsymbol{A}$, where we just enhance the difference in high-frequency part between $\boldsymbol{A}$_ and $\boldsymbol{A}$, while keep the low-frequency part same. As can be seen in figure 7 and 11 in appendix D.4.2, the learnt $\boldsymbol{A}$_ and $\boldsymbol{A}$ are actually a pair of "optimal contrastive pair" obeying GAME rule. To avoid misunderstanding, we will replace "optimal" with "effective" in revision.
>
> 5. > Unclear experimental description. For example, In Table 1, the authors do not state what dataset is used in the experiment; in Figure 3, what is the meaning of 'ACC'?
>
> **Response**: $\textbf{(1)}$ For the first question, actually we have stated that we use Cora for the results in Table 1. Please refer to Line 152-154, "With Eq. (2), we calculate the eigenvalues on $\textbf{Cora}$ after $\textbf{each augmentation}$, and plot their graph spectrums in Fig. 5. $\textbf{Simultaneously}$, we use the GCL framework in Section 3 to test the performances of $\textbf{these augmentations}$, and results are shown in Table 1.".
>
> $\textbf{(2)}$ For the second question, we also have clarified that we report accuracy in Figure 3. Please refer to Line 129-130, "The $\textbf{accuracy}$ is shown in Fig. 3.". In Figure 3, we use "ACC" as abbreviation of accuracy, which is a widely used abbreviation, such as page 285 in data mining textbook [1], table 4 in SimCLR [2], and section 6.2 of [3].
>
> [1] Han J, Pei J, Tong H. Data mining: concepts and techniques[M]. Morgan kaufmann, 2022.
>
> [2] Chen T, Kornblith S, Norouzi M, et al. A simple framework for contrastive learning of visual representations. ICML, 2020.
>
> [3] Zhu M, Wang X, Shi C, et al. Interpreting and unifying graph neural networks with an optimization framework. The Web Conference 2021.

---

> ### Author Response · Authors · 2022-08-02
> **Response (2/3)**
>
> 3. > Incomplete experiments. Apart from node classification, graph classification is one of the important tasks in graph contrastive learning. However, in this paper, the authors only show the results on node classification and ignore graph classification. In addition, in Table 2, why do not use MVGRL+SpCo (considering that MVGRL seems to be the best baseline used in this paper)?
>
> **Response**:  $\textbf{(1)}$ On the node classification and graph classification tasks, we agree that graph classification is one important task in graph analysis, however, it can be seen that conducting both of these two tasks is actually not a mandatory requirement in graph contrastive learning, for example, two representative baselines GCA and CCA-SSG where we followed.
> More importantly, in this paper, we mainly study topological augmentations for GCL, and propose GAME rule from $\textbf{graph spectral}$ domain. Some recent studies based on graph spectrum only focus on the node classifications, such as [1-3]. However, to our best knowledge, in traditional graph spectral field, there lacks theorem about which part of frequency components is useful for graph classification. Therefore, in this context, we firstly validate our model on node level tasks, and more exploration in graph level tasks is left for the future.
>
> $\textbf{(2)}$ Usually, the graph contrastive learning can be roughly divided into three categories based on their contrastive losses: BCE loss (DGI, MVGRL), InfoNCE loss (GRACE, GCA, GraphCL) and CCA loss (CCA-SSG). To comprehensively evaluate the performance of our model, we wish that our model is general to different categories, so we select one baseline from each category to integrate with SpCo, i.e., DGI, GRACE and CCA-SSG. Moreover, according to your comment, we further test SpCo with MVGRL, GCA, and GraphCL on Cora, where the results are shown in the following table. We can see that SpCo still outperforms base models. This further shows the effectiveness of our model, and we will open the source code.
>
> |       | split |  MVGRL   |     MVGRL+SpCo      |   GCA    |      GCA+SpCo       | GraphCL  |    GraphCL+SpCo     |
> | :---: | :---: | :------: | :-----------------: | :------: | :-----------------: | :------: | :-----------------: |
> | Ma-F1 |   5   | 76.2±0.5 | $\textbf{76.7±0.5}$ | 70.7±2.8 | $\textbf{72.7±1.8}$ | 75.1±0.4 | $\textbf{75.5±0.8}$ |
> | Ma-F1 |  10   | 78.4±0.8 | $\textbf{78.7±0.3}$ | 75.5±1.4 | $\textbf{76.7±1.4}$ | 77.8±0.7 | $\textbf{78.3±0.6}$ |
> | Ma-F1 |  20   | 81.5±0.5 | $\textbf{82.2±0.6}$ | 79.9±1.1 | $\textbf{80.2±0.5}$ | 80.7±0.9 | $\textbf{81.8±0.7}$ |
> | Mi-F1 |   5   | 77.0±0.7 | $\textbf{77.5±0.6}$ | 71.5±2.8 | $\textbf{73.4±1.7}$ | 76.4±0.3 | $\textbf{77.0±0.7}$ |
> | Mi-F1 |  10   | 79.4±0.8 | $\textbf{79.7±0.4}$ | 76.4±1.7 | $\textbf{77.8±1.2}$ | 79.0±0.8 | $\textbf{79.4±0.6}$ |
> | Mi-F1 |  20   | 82.8±0.4 | $\textbf{83.4±0.6}$ | 81.1±1.0 | $\textbf{81.6±0.6}$ | 82.3±0.9 | $\textbf{83.0±0.7}$ |
>
> [1] He M, Wei Z, Xu H. Bernnet: Learning arbitrary graph spectral filters via bernstein approximation. NIPS, 2021.
>
> [2] Wang X, Zhang M. How Powerful are Spectral Graph Neural Networks. ICML, 2022.
>
> [3] Bo D, Wang X, Shi C, et al. Beyond low-frequency information in graph convolutional networks. AAAI, 2021.

---

> ### Author Response · Authors · 2022-08-02
> **Response (1/3)**
>
> We sincerely thank the Reviewer for your reading and your correction of our typos. To answer your questions, we carefully clarify our motivation and contributions, and restate some details of paper. Additional experimental results are provided to supply our experiments.
>
> 1. > The notations in this paper is confusing. For example, in Line 80 and Line 81, ξ -> E, N -> $N$, $\textbf{A}$ -> $\boldsymbol{A}$.
>
> **Response**: Thanks for pointing this out. Here, N and $N$ mean the number of nodes, $\textbf{A}$ and $\boldsymbol{A}$ mean the adjacency matrix. Here, we do not find the usage of $\mathcal{E}$ in our paper, and we use $\xi$ to represent the set of edges. We will correct these typos.
>
> 2. > The proposed method has a serious limitation. In the paper, the authors fix one of the two views as the raw adjacency matrix, but do not discuss the two augmented views; in addition, some other augmentation operations (such as feature masking) are not discussed.
>
> **Response**: $\textbf{(1)}$ Please notice that the selection of certain views does not impact the correctness of GAME rule, implying that the results of GAME rule actually do not rely on the selection of different views. That's because GAME rule only indicates two contrasted graphs should have a smaller margin in low-frequency part and a larger margin in high-frequency part, without any limitation on the type of augmentation. For the convenience of analysis, as well as considering that most of augmentations are obtained from  the raw adjacency matrix $ \textbf{A}$, it is a natural setting that one view is fixed as $\textbf{A}$ and the other is an augmented one. In this case, we can identify the difference between augmentation and $\textbf{A}$, and calculate this difference from graph spectrum based on matrix perturbation formula (2) in Line 149. Finally, although we analyze between $\textbf{A}$ and its augmentation, our designed model demonstrates the generality of the GAME rule, where we propose a general plug-in, SpCo, based on this rule. SpCo can generally boost the performance of base models.
>
> As the general framework of GCL shown in figure 1(a), we can see that augmentations $\boldsymbol{V_1}$ and $\boldsymbol{V_2}$ are generated from original graph $\boldsymbol{A}$ . Therefore, if GAME rule is suitable for $\boldsymbol{A}$ and its augmentation, it will also be suitable for $\boldsymbol{V_1}$ and $\boldsymbol{V_2}$.  Derivation from $\boldsymbol{A}$ does not curb the generality of GAME rule. On the other hand, we show the diagram of our SpCo in figure 6. As shown in this figure, although we aim to generate $\boldsymbol{A}$_ from $\boldsymbol{A}$ obeying GAME rule, existing augmentations are still operated on $\boldsymbol{A}$ and $\boldsymbol{A}$_ to generate $\boldsymbol{V}_1$ and $\boldsymbol{V}_2$. In this case, the target model still receives two augmented views as you said. The improvement of performance in table 2 shows that the pipeline of SpCo can work and the GAME rule is effective.
>
> $\textbf{(2)}$ As for some other augmentation operations (such as feature masking), here our main motivation is to revisit the current graph contrastive learning from graph spectral theory, so it naturally connects to graph topology. Meanwhile, as can be seen in many literatures [1, 2], the graph topology augmentation is one of the most fundamental strategies in graph contrastive learning, therefore many different topology augmentation strategies are proposed, and building the relations among different topology augmentations is our main goal in this paper. While there is one mainstream augmentation designed for features in graph, i.e., feature masking, so we do not aim to explore the relation from only one feature masking strategy. Thus we mainly focus on analyzing topology augmentation in this paper, as clearly demonstrated in many different parts in this paper, e.g., Line 288. More discussions on feature augmentation can be left in the future.
>
> [1] Hassani K, Khasahmadi A H. Contrastive multi-view representation learning on graphs. ICML, 2020.
>
> [2] You Y, Chen T, Sui Y, et al. Graph contrastive learning with augmentations. NIPS, 2020.

---

> ### Author Response · Authors · 2022-08-09
> **Hope for your reply**
>
> Dear Reviewer ndDo:
>
> We thank you for taking the time to provide critical comments. We have provided detailed responses that we believe have covered your concerns. As this is the last day for discussion, we kindly remind you that could you check out our reply. We hope to further discuss with you whether or not your concerns have been addressed. Please let us know if you still have any unclear parts of our work.
>
> We are looking forward to your reply.
>
> Best,
>
> Paper5350 Authors.

---

### Official Review · Reviewer_Zdi6 · 2022-07-19

**Rating:** 8
**Confidence:** 4
**Soundness:** 3 good
**Presentation:** 3 good
**Contribution:** 3 good

**Summary:**

In this work, the high-frequency and low-frequency information are explored in the graph contrastive learning framework. The graph augmentation strategies are investigated from the spectral domain. A new spectral graph contrastive learning method is proposed based on the matrix perturbation theory. Experiments on five datasets show the model effectiveness.

**Questions:**

What are the specific parameter settings of compared baselines in the performance comparison section?

How is the model scalability of the new proposed SpCo component when integrating with the existing graph contrastive learning methods?


**Limitations:**

The detailed setting descriptions of compared baselines are missing in the evaluation section. For example, for the baseline DGI and GRACE, how the model depth is configured for performing the message passing. What is the graph scale in DGI to generate the graph-level representation for self-supervised learning task.

**Strengths And Weaknesses:**

Strengths:
1.The studied problem is very interesting and important for research works on graph contrastive learning.
2.The conducted experiments are comprehensive by demonstrating the effectiveness of the new proposed SpCo method in improving the graph representation performance.
3. Hyperparameter studies are provided to show the sensitivity of parameters.

Weaknesses:
1. The detailed parameter settings of compared baselines can be described to better understand the performance superiority of the new proposed SpCo method.

2. It would be better to perform analysis on the computational cost of the new proposed SpCo component in graph contrastive learning paradigm.

---

> ### Author Response · Authors · 2022-08-02
> **Response**
>
> We sincerely thank for your praise for our paper and valuable time spent on reviewing. To further address your issues, we display some important hyper-parameters for GCL baselines and discuss the computational cost of our SpCo as follows.
>
> 1. > What are the specific parameter settings of compared baselines in the performance comparison section?
>
> **Response**: We generally follow the settings in their original papers with carefully tune, and we report some important parameters of GCL baselines here.
>
> **DGI**
>
> | patience | lr    | l2_coef | hid_units            | drop_prob | nonlinearity |
> | -------- | ----- | ------- | -------------------- | --------- | ------------ |
> | 20       | 0.001 | 0       | 512 (256 for PubMed) | 0         | PRelu        |
>
> **MVGRL**
>
> | patience | lr    | l2_coef | hid_units | sample_size | batch_size |
> | -------- | ----- | ------- | --------- | ----------- | ---------- |
> | 20       | 0.001 | 0       | 512       | 2000        | 4          |
>
> **GRACE**
>
> |             | learning_rate | num_hidden | num_proj_hidden | activation | num_layers | augmentation                                           | tau  | num_epochs | weight_decay |
> | ----------- | ------------- | ---------- | --------------- | ---------- | ---------- | ------------------------------------------------------ | ---- | ---------- | ------------ |
> | Cora        | 5e-4          | 128        | 128             | ReLU       | 2          | drop_edge_rate: 0.2 / 0.4 drop_feature_rate: 0.3 / 0.4 | 0.4  | 200        | 1e-5         |
> | Citeseer    | 1e-3          | 256        | 256             | PReLU      | 2          | drop_edge_rate: 0.2 / 0.0 drop_feature_rate: 0.3 / 0.2 | 0.9  | 200        | 1e-5         |
> | BlogCatalog | 1e-3          | 256        | 256             | ReLU       | 2          | drop_edge_rate: 0.1 / 0.4 drop_feature_rate: 0.2 / 0.0 | 0.7  | 1000       | 1e-5         |
> | Flickr      | 1e-3          | 256        | 256             | ReLU       | 2          | drop_edge_rate: 0.4 / 0.1 drop_feature_rate: 0.0 / 0.2 | 0.7  | 1000       | 1e-5         |
> | PubMed      | 1e-3          | 256        | 256             | ReLU       | 2          | drop_edge_rate: 0.1 / 0.4 drop_feature_rate: 0.2 / 0.0 | 0.7  | 1500       | 1e-5         |
>
> **GCA**
>
> |             | learning_rate | num_hidden | num_proj_hidden | activation | augmentation                                           | tau  | num_epochs | drop_scheme |
> | ----------- | ------------- | ---------- | --------------- | ---------- | ------------------------------------------------------ | ---- | ---------- | ----------- |
> | Cora        | 0.01          | 256        | 64              | ReLU       | drop_edge_rate: 0.3 / 0.5 drop_feature_rate: 0.1 / 0.1 | 0.3  | 1000       | degree      |
> | Citeseer    | 0.01          | 256        | 64              | ReLU       | drop_edge_rate: 0.3 / 0.5 drop_feature_rate: 0.1 / 0.1 | 0.3  | 1000       | degree      |
> | BlogCatalog | 0.0005        | 256        | 256             | RReLU      | drop_edge_rate: 0.3 / 0.2 drop_feature_rate: 0.3 / 0.4 | 0.4  | 1000       | degree      |
> | Flickr      | 0.0005        | 256        | 256             | RReLU      | drop_edge_rate: 0.3 / 0.2 drop_feature_rate: 0.3 / 0.4 | 0.4  | 1000       | degree      |
> | PubMed      | 0.0005        | 256        | 256             | RReLU      | drop_edge_rate: 0.3 / 0.2 drop_feature_rate: 0.3 / 0.4 | 0.4  | 1000       | degree      |
>
> **GraphCL**
>
> | aug_type | drop_percent | patience | lr    | l2_coef | drop_prob | hid_units            |
> | -------- | ------------ | -------- | ----- | ------- | --------- | -------------------- |
> | degree   | 0.2          | 20       | 0.001 | 0       | 0         | 512 (256 for PubMed) |
>
> **CCA-SSG**
>
> |             | epochs | lambd | dfr  | der  | lr1  | wd1  | hid_dim | out_dim | n_layers |
> | ----------- | ------ | ----- | ---- | ---- | ---- | ---- | ------- | ------- | -------- |
> | Cora        | 50     | 1e-3  | 0.1  | 0.4  | 1e-3 | 0    | 512     | 512     | 2        |
> | Citeseer    | 20     | 5e-4  | 0.0  | 0.4  | 1e-3 | 0    | 512     | 512     | 1        |
> | BlogCatalog | 100    | 1e-3  | 0.3  | 0.5  | 1e-3 | 0    | 512     | 512     | 2        |
> | Flickr      | 100    | 1e-3  | 0.3  | 0.5  | 1e-3 | 0    | 512     | 512     | 2        |
> | PubMed      | 100    | 1e-3  | 0.3  | 0.5  | 1e-3 | 0    | 512     | 512     | 2        |
>
> 2. > How is the model scalability of the new proposed SpCo component when integrating with the existing graph contrastive learning methods?
>
> **Response**: With the additional SpCo, the computational cost of other existing GCL methods will not increase. In formula 10, we will make $\Delta_{\boldsymbol{A}}$ sparse by multiplying $\mathbb{S}$, which guarantees that $\boldsymbol{A}$ will be added by a sparse matrix. Also, the computation of $\Delta_{\boldsymbol{A}}$ is implemented on CPU, so this process will not account for any computational resource of GPU.

---

> > ### Comment · Reviewer_Zdi6 · 2022-08-09
> > **aknowledgement of author response**
> >
> > Thanks for the providing response. My comments have been well addressed. I would like to raise my rating score and vote for acceptance.

---

### Meta-Review · Area_Chair_okd3 · 2022-08-26

**Recommendation:** Accept
**Confidence:** Less certain

**Metareview:**

This paper had borderline reviews. The reviewers felt that the contributions were significant and that the connection between graph contrastive learning and spectral properties of the graph were valuable. Weaknesses included a focus on just node classification and some significant issues with presentation. The authors addressed many concerns presented and agreed to address the presentation issues. We hope that they will indeed do so in the final version of the paper, as the presentation issues would significantly limit the impact of the paper.

In our discussion one reviewer strongly advocated rejection, while three pushed for acceptance, putting the paper above bar.

**Award:**

No

---

### Decision · Program_Chairs · 2022-09-14

Accept